# An Examination of Perceptions among Black Women on Their Awareness of and Access to Pre-Exposure Prophylaxis (PrEP)

**DOI:** 10.3390/ijerph21081084

**Published:** 2024-08-16

**Authors:** Mandy J. Hill, Sarah Sapp, Shadawn McCants, Jeffrey Campbell, Akeria Taylor, Jamila K. Stockman, Diane Santa Maria

**Affiliations:** 1Department of Emergency Medicine, McGovern Medical School, University of Texas Health Science Center at Houston (UTHealth Houston), Houston, TX 77030, USA; sarahsapp96@gmail.com (S.S.); akeriataylor@yahoo.com (A.T.); 2Allies in Hope Houston, Houston, TX 77030, USA; shadawnmccants@gmail.com (S.M.); campbellj@aihhouston.org (J.C.); 3Department of Medicine, University of California, La Jolla, San Diego, CA 92093, USA; jstockman@health.ucsd.edu; 4Department of Research, Cizik School of Nursing, University of Texas Health Science Center at Houston (UTHealth Houston), Houston, TX 77030, USA; diane.m.santamaria@uth.tmc.edu

**Keywords:** pre-exposure prophylaxis (PrEP), cisgender black women, locus of control, facilitators to PrEP, barriers to PrEP

## Abstract

Cisgender Black women (CBW) experience 67% of new HIV diagnoses among women in the South. Progress toward ending the HIV epidemic requires researchers to explore perceptions of factors related to the decision to initiate pre-exposure prophylaxis (PrEP) among CBW. Qualitative methods were used to explore how social and structural constructs influence individual decisions to use PrEP among 20 CBW through focus groups. The thematic data analysis identified how facilitators and barriers to PrEP uptake aligned with an external locus of control (LOC) [e.g., media influences on understanding of PrEP] or an internal LOC (e.g., awareness of personal vulnerability to HIV). Several participants highlighted that their PrEP knowledge was rooted in an external LOC, such as media campaigns. A participant stated, ‘*But even with the commercial, it wasn’t representation for me.*’ Another participant described her personal HIV vulnerability in her sexual relationship as an internal LOC, stating, ‘*Not ignorance, it’s maybe just not accepting the true reality of this can be contracted even from someone that you believe that you trust.*’ Due to gaps in media marketing, healthcare providers should be aware that some female patients may perceive that PrEP is not for them. Provider-led sexual health discussions are urgently needed to bridge the gap between PrEP eligibility and initiation.

## 1. Introduction

Identifying the pathway to increasing pre-exposure prophylaxis (PrEP) initiation, defined as accepting an initial PrEP prescription [1], with cisgender (cis) Black women is a necessary component of the Ending the HIV Epidemic (EHE) initiative that seeks to reduce the number of new HIV infections in the United States (US) by 75% by 2025, and then by at least 90% by 2030, through four key strategies or pillars—diagnose, treat, prevent, and respond [2]. Black women account for nearly 60% of new HIV cases among US women while comprising less than 15% of the female population [3]. The reason for this disproportionate burden of HIV vulnerability is multifaceted and includes social and structural discrimination and biases that work together to keep cis Black women oppressed, thereby marginalizing their access to effective prevention tools that include PrEP. For example, PrEP prescription is lower for Black patients due to healthcare provider perception of higher-sex-risk behaviors. Additionally, factors such as homelessness, low socioeconomic status, and incarceration are associated with decreased PrEP uptake [4]. As a result, medical mistrust [5,6,7] and lived experiences of dehumanization within the healthcare system magnify the negative impact of systemic discrimination. Countering these systemic forces requires a mechanism that is built on the voices of cis Black women to readily engage cis Black women in a new behavior, specifically PrEP initiation.

PrEP is FDA approved with proven effectiveness at reducing the risk of HIV transmission through sex by 99% when taken as prescribed; it can be administered as a bi-monthly injection or an oral daily medication. However, less than 2% of eligible cis Black women take PrEP [8]. Healthcare providers, public health practitioners, and researchers are aware of the dire need to increase PrEP initiation with cis Black women; however, the paucity of funded interventional research focused on effectively linking cis Black women to PrEP initiation persists. No significant decrease in HIV rates or increase in PrEP uptake has yet been observed [9], thereby magnifying this glaring unmet need.

Bridging the gap between healthcare providers extending access to PrEP during clinical visits and eligible cis Black women being ready to take PrEP requires a growing awareness of why the gap exists in the first place. Survey data demonstrate that some healthcare providers in Texas are not comfortable prescribing PrEP [10]. A single-arm quantitative pilot study among women in Texas demonstrated interest in and willingness to use PrEP, but they were not current PrEP users [11]. Thus, when women are supported to access PrEP, it is imperative that they are received well by knowledgeable healthcare providers who can and will offer PrEP. With respect to healthcare providers, positive interactions with informed and culturally competent clinical staff [5] have been noted as a facilitator for PrEP initiation. Achieving equitable access to PrEP requires consideration of the healthcare system and environment, specifically access to a discreet and convenient clinic, insurance coverage, and availability of comprehensive services in specified locations to assist clients with PrEP access. Lastly, receiving navigation services, in addition to the aforementioned services, was perceived as a healthcare benefit (64% of clients receiving services said they initiated PrEP) and was listed as a facilitator of PrEP initiation [5].

Some research studies have reported both facilitators and barriers to PrEP initiation among cis Black women. Barriers identified included limited PrEP awareness, low perceived HIV risk, concerns about side effects, concerns about costs, limited marketing, and distrust in the healthcare system [12]. Reported facilitators of PrEP initiation with cis Black women include women’s empowerment and advocacy, awareness of HIV risk (specifically with partner disclosure [5,13,14]), provision of education that is specifically about PrEP, and the positive influence of PrEP-engaged women’s testimonials [5,12]. In a study (2017–2019) where 26% of 565 cis women in New York City initiated PrEP, Latina women (29.7%) and Black women (26.1%) were more likely to initiate PrEP when compared to White women (16.3%) [5]. Findings demonstrated that PrEP initiation was related to PrEP awareness, low income, unstable housing, receipt of navigation services, and reports of non-injection substance use and/or a recent sexual relationship with an HIV-positive partner [5]. The facilitators and barriers to PrEP noted here are diverse. It has been posited that the use of vlogs for sexual health communication with cis Black women can be designed and implemented in innovative intervention strategies that connect with this audience in a way that aligns with the communication norms of society and communities [15]. Consequently, approaches to increasing PrEP initiation with cis Black women should focus on communicating information in a way that amplifies a uniform health message of why PrEP initiation is an important and necessary behavior change.

In relation to health communication about PrEP to the broader community, cis Black women shared a limited awareness of PrEP, which is exacerbated by the lack of cis Black women-specific marketing [12]. Recognition of competing priorities (i.e., housing instability and food insecurity) as a barrier to PrEP initiation and/or readiness elucidates how more immediate needs for cis Black women take precedence over the priority of engaging in HIV prevention behaviors [5]. When researchers carefully consider how a person’s way of life can influence how preventive healthcare is prioritized, program developers and interventionists have the opportunity to develop and test person-centered health messages that can validate the human experience and enhance skills in prioritizing health amidst more pressing lifestyle needs.

The study team recognizes that literary evidence evaluating implementation intentions of PrEP with cis Black women is a necessary first step towards the development and implementation of behavioral interventions that can effectively increase PrEP initiation. As such, we led a qualitative study to explore facilitators and barriers that influence the decision for PrEP initiation with cis Black women. Qualitative findings here informed the next phase of the research, which was the development of health communication material from the perspective of cis Black women to inform the content of a video-log-based intervention to enhance access to PrEP, culminating in the final phase to pilot test the vlogs and assess whether they facilitated behavior change, specifically an increase in efforts to access PrEP over a brief follow-up period.

## 2. Materials and Methods

This is a qualitative observational study using focus group (Appendix A) discussions in a virtual environment as the communication medium to explore factors that may influence PrEP initiation among a cohort of 20 cis Black women. The study was reviewed by an institutional review board and was approved (HSC-MS-21-0419).

We recruited a purposive sample of 20 PrEP-eligible cis Black women through convenience sampling using electronic flyers (e-flyers) that were disseminated through social media channels of academic and community partners as well as in-person recruitment through social networks of community-based agencies and clinics. Eligible participants were assigned female sex at birth, were 18 years or older, had been sexually active with an opposite sex partner within the past 6 months, were fluent in English, and had access to a phone or internet. Cis Black women who were not able to physically provide informed consent due to cognitive impairment or psychological distress, were unable or unwilling to meet study requirements, were ineligible for PrEP (i.e., known to be HIV-positive), or were already on PrEP were not eligible to participate. The e-flyers contained a QR code that linked potentially eligible participants to a Qualtrics survey that assessed their eligibility for study participation.

Cis Black women who were not able to provide informed consent or were ineligible for PrEP were excluded. Those enrolled provided contact information (i.e., name, email, and phone number) so that a research coordinator (RC) could explain the study to them, send them an electronic consent form using DocuSign software v9.4, and send them a weblink to the online scheduler where they could choose a date and time for participation in a focus group discussion (FGD). Once the session was scheduled, participants completed a baseline survey in Qualtrics to provide their contact information (i.e., email and phone number) for use by the RC to follow up with them on the next business day to explain the details of study participation.

The FGD guide included open-ended questions about the influences that culture, race, and gender have on PrEP use with cis Black women. The theoretical framework to justify these influences on PrEP use is rooted in the Theory of Gender and Power and The Sexual Script theory. The Theory of Gender and Power describes how societal norms influence gender-based inequities in a culture where males are socialized and supported in controlling decisions, which inform relationship dynamics and the way partners communicate and make sexual decisions [16]. The Sexual Script Theory explores the ways culture shapes perception and expressions of sexual behavior in ways that individuals and partners deem appropriate and socially acceptable. The script becomes the roadmap of how sexual cues are interpreted and the behavior that follows [16]. The FGD guide explored elements of the Theory of Gender and Power and The Sexual Script theory to identify drivers of the decision to access and use PrEP in order to (1) prevent HIV transmission and (2) protect one’s sexual health and wellness [16,17,18]. During the scheduled FGD, the RC ensured participant names were replaced with a study identification number to maintain anonymity. The RC then began the FGD, explained the study, facilitated the discussion using the FGD guide that was developed by the investigative team, and audio-recorded the session with transcription using a videoconferencing software (i.e., Microsoft Teams v1.7 or Zoom v6). Participants received a physical gift card valued at USD 50.00 via USPS mail. The study team conducted a total of 7 focus groups, coded 7 FGD transcripts, compared codes, discussed differences, and reached a consensus using a constant comparative analytic approach to revise the final codebook [19]. Coded data were analyzed for emergent categories and themes [20], which guided content development for vlogs for cis Black women in the next phase of this study.

The presentation of themes was organized by locus of control as either internal or external. The theory of planned behavior [21], which includes the theoretical concept of perceived behavioral control, can be applied to PrEP initiation as a new prevention behavior. A locus of control, which is a personality trait measure used to study personality and characteristics, describes the degree of control a person perceives themselves as having to perform the behavior itself [22], which in this case is the behavior of taking PrEP for HIV prevention. A locus of control can be internal or external. Individuals with an internal locus of control believe that they can control events and outcomes in their lives; however, individuals with an external locus of control believe that their life is governed by external forces such as other people and chance [23,24,25,26,27,28]. Themes are organized based on locus of control as a predictor of PrEP initiation among cis Black women.

### Data Analysis

The screening tool garnered 241 responses. We assessed demographic data at the individual level regarding age, biological sex, income, insurance status, and sexual orientation. Most respondents were female, young adults aged 18–29 years, residents of the largest, most diverse, and populous county in the South, engaged in condomless sex with a male partner within the last 12 months, and presumed HIV-negative (see Table 1).

The three-member coding team co-created an initial codebook using one transcript during a series of virtual meetings using Microsoft Teams v 1.7 software. The team lead has robust experience analyzing qualitative data using thematic content analysis [29,30,31,32]. The analysis team compared coding, discussed differences, and reached a consensus using a constant comparative analytic approach to revise the final codebook. Once the codebook was developed, it was used when each coder completed their coding independently. New codes were added only when new codes emerged. The updated codebook, with the addition of new codes, was reviewed by the team and duplicate codes were collapsed. As an example, relative to the same quote, one coder created a new code labeled as ‘preference in having a female provider’ and the second coder labeled the same quote as ‘perceived comfort in having a female provider’. During the final revision of the codebook, these two codes were collapsed as ‘preference in having a female provider’. Then, coded data were analyzed for emergent categories and themes. Only two of the three assigned coders completed the coding assignment. Using the statistical software package IBM SPSS v 26.0, the team calculated a Kappa statistic to assess inter-rater agreement between coders.

The inter-rater reliability was assessed based on a random selection from a review of 50% of codes (across 514 individual codes of data) with 1055 total codes, resulting in an agreement between coders 95.8% of the time, a Kappa statistic of −0.002, and SE = 0.001 between coder 1 and coder 2 across 99 valid cases. This Kappa statistic infers that there was little to no difference between the observations of the coders and chance alone. The team generated a report and conducted member checking with community partners during professional meetings nationally, roundtable discussions, and presentations of preliminary data and concepts across the local sexual health community. Inter-rater agreement was assessed using a Kappa Statistic in SPSS v.26.0 across all codes identified. The negative Kappa statistic across all three coders suggests no agreement. In response to this result, the study team conducted an assessment of 99 codes with quotes across both coders. After completing the process of collapsing codes, disagreements between both coders were verified only three times across 99 quotes via eyeball tests.

## 3. Results

### 3.1. Thematic Findings

The full thematic analysis of all focus group and interview transcripts garnered 1217 codes. All codes in this qualitative analysis were related to participating cis Black women making the decision of whether or not to take PrEP. A total of 17 themes were identified with the thematic analysis. Codes are stratified based on whether the team perceived the code as stemming from an external or internal locus of control. When the participants’ responses aligned with an event or view that the participant perceives is outside of their control and drives their actions and outcomes, these codes were assigned to the theme ‘external locus of control’. However, when participants presented information where they believed that they were in control of their own abilities, actions, or mistakes, these codes were assigned to the theme ‘internal locus of control’. The codes with the highest frequency are highlighted here. Themes and associated codes are presented in Table 2.

#### 3.1.1. External Locus of Control

Subthemes for the external locus of control relative to the decision to take PrEP were (1) self-advocacy with providers about health, (2) media influences on understanding of PrEP, (3) comfort with provider–patient relationships, (4) provider engagement in sexual health of cis Black female patients, (5) awareness of peers with HIV and/or taking PrEP, and (6) PrEP-related concerns (Table 2).

##### Theme 1: Self-Advocacy with Providers about Health

The codes that comprised Theme 1 describe the experience that cis Black women had with healthcare providers when an external environment was created where they felt empowered or disempowered to self-advocate for their health. Together, these codes describe the external elements that contributed to their comfort with self-advocating for their health in that environment, which include racial and/or gender concordance, or external factors that detracted from their willingness to advocate for their health, which included cues that their provider’s knowledge was outdated or experiences with failure of their healthcare provider to broker sexual health discussions.

Code: Preference in having a Black provider

This code brings together quoted text whereby participants described their rationale for preferring a healthcare provider who had the same racial identity as them, meaning that they shared the same external identity in the world. Based on the quotes for this code, the participants described this racial concordance as an external factor that increased the likelihood that their healthcare provider would perceive them as fully human at the initial encounter, which they described as giving them access to a higher quality of care with an opportunity for their voices to be heard and their concerns to be fully considered.

When asked about preferences when choosing a healthcare provider when speaking about sexual health, one participant stated,


*‘Yeah, I would want somebody Black because sometimes the other race don’t know what I’m talking about. Yeah, that matters. Yeah, I’m sorry, that matters.’*


In the instances where this code was used, participants expressed that racial concordance between provider and patient created a level of mutual understanding that may not be present when healthcare providers are of the White race. In some cases, the gender and race of the provider as a White male are described.

2.Code: Preference in having a female provider

Similar to the perceived benefits of racial concordance, some participants preferred concordance at the biological sex or gender identification level. The quotes under this code described the sameness of presentation in society as female between themselves and their healthcare providers as an external factor that would likely increase their chances of compassion and access to humane treatment.

Similarly, when asked about preferences with the characteristics of a healthcare provider concerning sexual health, a participant in focus group 5 stated,


*‘I actually just had an appointment with them yesterday morning and I feel comfortable. They’re all women. So, ofcourse, I feel more comfortable talking to a woman than a man in my opinion. And they don’t tell me anything wrong. Any questions I have, any concerns, they always answer them. And then if they see something that’s not looking too good, they don’t hesitate to call me or text me and stuff like that. So, I’m very comfortable with my healthcare provider.’*


Here, we see a contrast in feelings of comfort when there is gender-based concordance between the provider and the patient, where participants expressed a higher level of comfort when providers are female.

##### Theme 2: Media Influences on Understanding of PrEP

Codes describing the experience of cis Black women with media messaging regarding PrEP revealed diverse perceptions about how PrEP is prioritized for Black women, messaging about the LGBTQ+ community, perceptions around gender-based messaging, and the relatability of messaging and feeling represented in the media. Together, this theme illuminated gaps that cis Black women described related to media influences on their understanding of PrEP as an HIV prevention option for them.

Code: TV Media messaging about PrEP

This is the most frequently coded item amongst all codes in this study with a total of 60 uses. Within this code, participants described a diversity of perspectives gleaned from witnessing media messaging about PrEP on television (TV). Many of the perspectives were shaped by who was presented in the commercial, which informed how cis Black women determined who they presumed was eligible for PrEP.

When the focus group facilitator asked participants, ‘Who knows about PrEP? Who has heard of PrEP? Who knows about PrEP?’, participants shared diverse thoughts. Most of their perceptions were gleaned from watching commercials about PrEP. One participant in focus group 5 stated,


*‘Actually, my first time actually seeing it was on How to Get Away with Murder. And I’m going to be honest, even as it may sound, I did not know PrEP medication was real. I thought that was just something that they put on TV.’*


In this quote, the participant acknowledges seeing the message and understanding that PrEP is a medication, but she questions the authenticity of the message and the realness of the medication.

In response to the question, ‘When you see advertisements and marketing about PrEP on different media outlets, how do you feel PrEP is portrayed to you?’ during focus group 3, a participant described her perspective in this way,


*‘I guess for if we’re talking about video media, it would definitely need to be people who represent the community that I represent, what they display right now is just LGBTQ, so it doesn’t gravitate at least what the commercials that I’ve witnessed.’*


Her statement reflects her observation that the media targets a segment of society that does not represent her, which she described as creating a disconnect and a presumption that the product presented is not relevant to her lived experience.

2.Code: Perception that cis Black/African American women are not prioritized in TV media messaging about PrEP

Participants described that the failure to adequately represent cis Black women in TV media messaging about PrEP externally in society seemed to inform and shape internal messages at the individual and group level with cis Black women that PrEP was not for them.

The facilitator of focus group 3 asked participants who had previously described what they knew about PrEP to share with fellow participants what they knew and what they did not know. One participant stated,


*‘But even with the commercial, it wasn’t representation for me. And so, it was like, “Oh, okay, so they’re trying to really get it for the LGBTQ community.” There was not any partners within the commercial that was heterosexual. So, I was like, “Oh, maybe this is not for me”.’*


This statement shows a connection between failure to see oneself in a commercial and deducing whether or not the content shared is applicable to one’s personal circumstance. The participant stated that her attention to the commercial was fleeting, as she felt the content did not relate to her as it would if it were related to a condition that she actually has to manage (i.e., a chronic health condition).

##### Theme 3: Comfort with Provider–Patient Interactions

Participants thoroughly described their interactions with healthcare providers and their feelings about those interactions. The codes within this theme describe the diversity of external contributors to internal comfort or discomfort when cis Black women engage their healthcare providers in health communication.

Code: Perceived comfort with discussing sexual health with provider

This code intends to describe how a provider can influence the way a cis Black female patient feels when discussing their own sexual health. The influence can be positive or negative and depends on the external stimuli of the provider.

The facilitator of focus group 5 asked participants 2 and 3 about their comfort with having sexual health conversations and testing with their providers. Participant 2 stated,


*‘I’m very comfortable. I’ve been going to my provider for a while, but I think even if I hadn’t been going to my provider for a while, because I have awareness of HIPAA and privacy and all of that, that helps me to increase my comfort level in knowing that anything that I need to discuss, this is where I should be able to do it without any judgment or without anybody knowing what it is that I want to discuss.’*


In this response, it is clear that participant 2 is keenly aware of her rights to privacy as a patient and has related expectations of the provider to maintain privacy-related matters.

2.Code: Communicating openly with your provider is a part of your health

The ability to communicate openly with a healthcare provider was described as a response to the emotional safety that some cis Black women described when certain preferred provider characteristics were present.

During focus group 3, participant 2 shared a conversation that she had with a provider where she describes a link between honesty and perceiving the provider as open to communication. She stated,


*‘In college, I have thought, because I had unprotected sex, I really thought I had HIV and I started having symptoms that you would think I was dying because of how I was thinking. So, I choose to only have sex with one person. So, I think because I responded that way, she didn’t further ask questions. But I think if I would have been honest, if it was the case in telling her that I was having sex with several individuals or I was having unprotected sex, it could have become a greater conversation, she seemed open.’*


In this quote, we see that participant 2 gave her provider a socially desirable response and presented a false case of being in a monogamous relationship although she admitted to the facilitator that she was engaging in condomless sex with multiple partners.

##### Theme 4: Provider Engagement in Sexual Health of Cis Black Female Patients

This theme brings together the codes that specifically convey the experiences of cis Black female patients when engaging providers in HIV/STI testing, sexual health discussions, and conversations about PrEP. Participants conveyed their psychosocial experiences of trust and perceived assumptions by healthcare providers.

Code: Experience with provider offering HIV testing

The responses of participants who described their experience with being offered an HIV test by their provider are assessed. The varied external factors of the experience, such as the tone set by the provider, the level of engagement, and the perception of the provider’s genuine interest and investment in the participant’s sexual health and well-being, are discussed.

During focus group 5, the facilitator describes PrEP as an option that is available to cis Black women. She then asked participants whether they have conversations with their providers about sex, sexual health, and/or HIV testing. Participant 1 stated,


*‘‘I go to a male OB-GYN doctor, and I kind of regret that, but I’ve had him for so many years. It’s just like you get used to him, you just go just for the checkup. But he doesn’t ask any questions. And definitely he just says, okay, this month we’re going to check for STDs, whatever, and whatever. And that’s pretty much it. And sometimes I get offended because I’m like, why are you checking for that? I didn’t ask you to check for that.’*


The participant shares a level of discomfort associated with the provider offering STD testing without the preface of a conversation to determine eligibility.

2.Code: Willingness to engage provider in HIV testing

As part of the theme assessing provider engagement in the sexual health of cis Black female patients, this study explored this code to gauge the initiation of sexual health discussions with providers by cis Black women in a clinical setting.

During focus group 3, after some time and discussion about provider–patient interactions, the facilitator specifically asked if participants had asked their provider for an HIV test or whether they recommended it to them. Participant 4 stated,


*‘I think I’ve done both. I think they’ve asked and then I think I’ve also asked them to run one.’*


Participant 4 offered a response that provides evidence of cis Black women leading sexual health discussions with providers and engaging in provider-led discussions that include HIV testing.

##### Theme 5: Awareness of Peers with HIV and/or Taking PrEP

This theme grouped together codes that describe the participants’ awareness of their peers’ experiences with PrEP and HIV. It also explored their curiosity about learning more and their willingness to embark upon open communication about PrEP.

Code: Awareness of peers on PrEP

Participants described their knowledge of their referent others and peers who shared that they were taking PrEP.

During focus group 6, the facilitator asked participants if they knew anyone that is on PrEP or if they took PrEP or considered taking PrEP. Participant 1 stated,


*‘I only know one person that is on PrEP and he’s actually gone through a couple of PrEPs. And the only thing I know is that it, well, one of them, it lessened his alcohol tolerance. We were having a drink and he was like, “Yeah girl, I need to be real with my doctor. And so, she know I be out here.” So, she was like, “Hey, we’re going to try something else that better aligns with your lifestyle.” And in that moment, I was like, man, that’s what’s up. But I also felt like why can’t I have that same relationship with my doctor where I could feel comfortable to be like, “Girl, yes I drink and yes, I be throwing it back.”, if something that goes with both of those things. But yeah, I just know that one person.’*


The example of a person on PrEP by the participant illustrated the person’s provider–patient relationship and their comfort and willingness to engage in discussions about sexual and social behaviors.

2.Code: Awareness of peers with HIV prompts willingness to learn more

This code assessed whether the external influence of peers normalizing an HIV-positive status had an influence on HIV-negative cis Black women in terms of their openness and willingness to learn more about HIV. The presumption here by participants is that when a person within your own social network is living with an infectious disease that is stigmatizing in society, the external stigma may be disempowered and the personal value of that individual may facilitate a genuine curiosity to learn about their lived experience.

During focus group 4, participant 2 engaged in an ongoing dialogue that was prompted by the facilitator asking about ways heterosexual cis Black women can be made to feel included in PrEP media marketing. One participant stated,


*‘I just think that realistically, because I always say, with diseases, if you don’t know anybody personally, it goes over your head. So, for us, had I not met two amazing people, one sitting here in this room, [de-identified name}, who had it, I would’ve never even gave HIV, or anything like that a second thought.’*


The participant described having an enhanced awareness of HIV by engaging someone with an HIV diagnosis. She described that attention to the topic was connected to knowing someone with the condition. The absence of a living example was linked to a dismissal of HIV and PrEP-related information.

##### Theme 6: PrEP-Related Concerns

The codes grouped together under this theme describe concerns about PrEP and external barriers that may discourage PrEP use, including the administration route of PrEP, costs, and personal readiness to use PrEP to prevent HIV.

Code: Concern with cost and insurance coverage for PrEP

Although the presentation of PrEP as a highly effective prevention modality seemed to appeal to participants, access to it requires financial means, affordability, insurance coverage, and/or awareness of and access to information for co-pay assistance. This code describes how participants expressed the potential financial burden of PrEP, an external factor that can serve as a limiting barrier to PrEP access.

During focus group 3, the facilitator asked participants about the reasons why a woman would choose not to use PrEP. Participant 2 stated,


*‘I’m very… I don’t like taking medication just period. So, for me it would be weighing the cost, risk, benefit if I’m really willing to take this pill every day. But I am now just learning, I believe this month about the injection, but I also have fears of injection.’*


In this response, we see an aversion to taking medication in general, which seems to thwart willingness to PrEP, in addition to gaps in knowledge relative to the costs, risks, and benefits of the daily medication.

2.Code: Concerns for PrEP injectable

The varied modalities of PrEP administration were discussed; however, this code specifically refers to PrEP delivery as an injection and the concerns that the participants raised around receiving an injection for HIV prevention. As is articulated in the quote below, the concern around the injection is not simply fear of an external needle but includes the historical social injustices that Black people have experienced through injections presented as a health benefit when physical harm has been done.

During focus group 3, participant 2 raised concerns about PrEP’s side effects. She stated,


*‘I believe this month about the injection, but I also have fears of injection and I think that kind of goes back to medical malpractice from back in the day when Black people were being given injections and it wasn’t what it said it was. And it’s the concern about side effects too.’*


The correlation between current concerns with injectable medication is linked to historical traumas to the Black community that fuel medical mistrust today with injections. This concern or fear is linked to concerns about the side effects of the medication. Together, these serve as barriers to PrEP initiation with cis Black women.

#### 3.1.2. Internal Locus of Control

Subthemes for the internal locus of control relative to the decision to take PrEP were (1) perceived PrEP knowledge, (2) PrEP willingness, (3) knowledge of PrEP as a prevention tool, (4) awareness of sexual risk, (5) perceived facilitators of the decision to use PrEP, (6) factors that inform the choice for or against PrEP use among cis Black women, (7) willingness for self-advocacy with healthcare provider, (8) willingness to educate others on PrEP, (9) acceptance of PrEP as approved, (10) awareness of HIV risk with drug use, and (11) perceived HIV knowledge (Table 3).

##### Theme 1: Perceived PrEP Knowledge

This theme describes the internal knowledge that participants believe they have about PrEP. In many cases, the basis of the knowledge is not factual or scientifically based. Yet, the influence of the perceived knowledge on the decision to take PrEP or support PrEP uptake is profound.

Code: Perception that PrEP is not for heterosexual people

Often guided by external influences such as media messaging, the internal belief that PrEP is not for heterosexual people described by cis Black women in this study illustrates a prevalent self-exclusion from an effective HIV prevention medication among an eligible population, as this code was the most frequently used code within this theme (*n* = 28).

In focus group 4, the facilitator specifically referred to participant 1 and referenced an earlier comment where participant 1 stated she felt that commercials and marketing for PrEP were geared towards homosexual individuals and people of trans experience. The facilitator then inquired with all participants about their thoughts on the marketing and who they feel the marketing is targeted to, whereby participant 1 stated,


*‘I was going to say, the reason why I say that is because I typically--I mean I’m not trying to be funny. See it when I’m watching shows like “Pose” and stuff like that where those shows are catered to that community. That’s usually when I would see that.*



*Or if it’s an episode of a show I’m watching and you have a gay couple on that show, that’s when I’m seeing it. To me, it’s Blacks, or Black females, I’m thinking maybe they should have it on more networks where people are watching so why is it not on …or other places, that’s the target.*



*I mean, I don’t know if they are showing it on those channels, but I’m just saying. And then in the commercials, like I said, I never knew that it was for straight. Because it’s not any straight couples on the commercials. It’s two men, but the lady hosting the commercial is transgender. So, I mean, I never knew that was…’*


Participant 1 provided the evidence that informed her conclusion that PrEP is not for heterosexual people based on the content shared and the communities targeted by PrEP commercials and marketing.

2.Code: Perception that PrEP is for gay men

Similar to the determination of who PrEP is not for, the deductive reasoning of whom PrEP is for based on observations of media messaging becomes governing information that influences decisions, perceptions, and beliefs. The perception that PrEP is for gay men was readily discussed by participants and was mentioned 24 times.

During focus group 6, the facilitator asked participants to describe their thoughts on why women would not take PrEP. Participant 1 stated,


*‘And then I could also see Black women just genuinely not taking it because they see it as the gay man’s drug. And so, if I take it, that means I am admitting to the world that either I am attracted to or have sexual relations with gay men, trans men, bisexual men, and what that would look like in the community if people knew that about me.’*


This response reveals two separate factors that may influence reasons why women would not take PrEP, including the perception of PrEP as a ‘gay man’s drug’ and fear of being perceived as having sex with gay, trans, or bisexual men by community members.

3.Code: Perception that PrEP eligibility is based on sexual orientation

The quoted text under this code highlights the perceptions gleaned from the messages and images purported by the media that become internal beliefs about who is eligible for PrEP. In this code, cis Black women linked PrEP eligibility to sexual orientation, specifically a link between PrEP eligibility and the LGBTQ community.

During focus group 4, the facilitator provided a foundation of knowledge on health inequities in HIV incidence to cis Black women. She then asked participants if they know what PrEP is and if they had ever heard of it. Participant 1 stated,


*‘Of course, I’ve seen the commercials, but I thought it was mostly targeted towards homosexuality, just based on the people that are in the commercials. I’ve never seen a straight couple. It’s usually transgender or two males.’*


Similar to other comments about PrEP being for members of the LGBTQ community, we see here that the deductive reasoning is that PrEP is not for heterosexual individuals if commercials are presenting individuals and couples who are seemingly of the LGBTQ community.

##### Theme 2: PrEP Willingness

Willingness to embark upon a new behavior can be motivated by internal and external drivers. The codes described here under the theme of PrEP willingness describe some of the internal fears and reasons that either facilitated or thwarted willingness for PrEP initiation among cis Black women in this study.

Code: Stigma around PrEP uptake

As it pertains to the stigma around taking PrEP, some participants described a contrast between generational perceptions, where preventative medications were perceived as permission to engage in sex freely, versus current perceptions where the medication is a means for self-protection. In other quotes, the stigma around PrEP uptake related to engagement in sexual practices that were historically described as ‘high-risk’.

The facilitator in focus group 4 asked participants to describe some of the reasons why women would choose PrEP. Participant 1 stated,


*‘But I think it comes with the stigmatism, like even when our parents used to struggle with the, “Well, if I give my child birth control, I’m giving them permission to just be free and have sex and no consequences because they’re on birth control.” So, someone can look at… PrEP like that, like, “Okay, if I’m taking this and I’m out here, I don’t have to use any other protection. Hey, I’m not going to get HIV.’*


In her response, participant 1 made a correlation between the freedom that PrEP can induce to engage in condomless sex and the perceived freedom of parents when they encourage their child to take birth control.

2.Code: Reasons women might be hesitant towards taking PrEP

After exploring some of the external factors that served as barriers to taking PrEP, the coding team discerned a pattern of quoted text describing the internal reasons why cis Black women may be hesitant to take PrEP.

During focus group 6, the facilitator asked participants to describe some reasons why women would choose not to take PrEP. Participant 2 stated,


*‘Because they don’t think it’s for them in general. So, I don’t have it. So why would I take something that I don’t need? I feel like a lot of people too in general have issues with taking medicine or putting medicine in their body in any type of form. It doesn’t matter what it is. There are so many people that are against so many vaccines and this, that and the other, that will have a huge problem with having to take something else that is really not required.’*


Participant 2 illustrates the polarity in perceptions of biomedical interventions for prevention when the social standard is to take medication for a diagnosed illness. The hesitancy, in her purview, is rooted in her internal value system, whereby her stance is not to take a medication that is not needed. Out of the quotes under this code, 57% referred to vaccine hesitancy with references to the COVID-19 vaccine and health communication by anti-vaxxers.

##### Theme 3: Knowledge of PrEP as a Prevention Tool

This theme describes the understanding of cis Black women in this study relative to the function of PrEP as a prevention tool that can be used to serve them and their sexual health needs.

Code: Perception that PrEP offers safety

This code stemmed from dialogue whereby cis Black women articulated their thoughts around the personal and communal safety that PrEP uptake could offer them. They described a realization of a phantom control over the sexual decisions of their partners, which they explained as enhancing their self-awareness of a need for additional tools to secure the safety of their own sexual health.

The facilitator in focus group 3 asked participants about their thoughts on the word empowerment as it relates to access to PrEP. Participant 2 stated,


*‘I absolutely think it would. For me, it would be more so if I knew I was having sex with multiple people, but then also you have to rethink that too. But I do think that it would be empowering to know that I have control over not contracting this condition.’*


Participant 2 described that she connected the feeling of empowerment to the control she had, through PrEP, to prevent herself from contracting HIV.

2.Code: PrEP empowers ability to control personal sexual health

This code reflects quoted text whereby PrEP is yet another tool for cis women to exercise bodily autonomy. In some cases, such as the quoted text below, PrEP is compared to birth control where a cis woman can choose to stave off an unplanned pregnancy. PrEP is perceived by some cis Black women in this study as yet another tool for women to have decision-making power relative to their own sexual health.

Similarly, during focus group 3, when the facilitator asked about empowerment relative to access to PrEP, participant 4 stated,


*‘I think it will. I definitely believe you’re now able to take hold on exactly what is going to happen with your body. You don’t have to rely on a condom or a STD test to determine, you can actually do take the precautions to say, “Hey, you know what? I took this pill or I took this shot so I know I’m doing what I need to do to protect myself.” So I think that’s something that comes to mind, just like if I would, that’s what my IUD, I’m protecting myself to not get pregnant. I should also now include this into my regimen of my health. And so, I think that it does allow for me to control if I want to have a baby as well as if I want to be exposed to someone, to something out there that possibly I do not want to have in my life at this time.’*


Similarly to how participant 2 describes the feelings of safety that PrEP offers her, she also describes it as a source of empowerment to control her ability to prevent herself from contracting HIV. Participant 4 perceives PrEP as a pathway to controlling what happens to one’s body. She also describes PrEP access as increasing options for self-protection beyond a condom or STD test. She also made a parallel connection between PrEP to HIV prevention and an IUD to pregnancy prevention.

##### Theme 4: Awareness of Sexual Risk

The responses from participants illuminated a palpable awareness of risks to sexual health through the decision to engage in sex with a male partner. A heightened awareness of sexual risk was conveyed when engaging with a new partner. This particular theme in this work has significant contributive value to the body of literature, whereby low perceived risk is a pervasive norm throughout the sexual health literature relative to cis Black women. This theme purports that this particular paradigm may be changing.

Code: Awareness of HIV risk within sexual relationships

Participants described an acceptance of reality, whereby in the presence of trust in an established and seemingly monogamous relationship, a risk of HIV is still present because of the inability to completely attest to the sexual behaviors of a sexual partner.

During focus group 3, the facilitator led a conversation on reasons why cis Black women may reject PrEP. Participant 2 described the possibility of being at risk for HIV whether married or not. This participant actually referenced a case where her best friend contracted HIV within marriage. As a follow-up, participant 2 then stated,


*‘Not ignorance, it’s maybe just not accepting the true reality of this can be contracted even from someone that you believe that you trust.’*


This quote reflects an acceptance that a risk of contracting HIV is present in spite of the trust that a person may feel within the relationship dynamic.

2.Code: Perception of elevated HIV risk with new partner

Participants described perceptions that the HIV risk with a new sexual partner is higher than the risk with an established partner due to the quantity of unknown information about their norms, their past, their behaviors, etc.

During focus group 6, the facilitator directed a question towards participant 2 in reference to a comment by participant 1, which was an inquiry about what would prompt her to get an HIV test. Participant 2 stated,


*‘Guess messing with someone new, a new situationship or whatever I’m doing. I’m definitely going to get tested.’*


This quote references the prompt to get tested when engaging in an uncommitted relationship with a new sex partner.

##### Theme 5: Perceived Facilitators of the Decision to Use PrEP

This theme brings together the quoted text describing how cis Black women in this study perceive factors that will encourage or motivate their decision to use PrEP. They described a diversity of factors that included the varied ways PrEP can be taken, increasing awareness that PrEP exists, and offering education about PrEP, as well as sharing that PrEP does not have to negatively impact the intimacy of sexual relationships.

Code: Perception that varied PrEP messaging can bring awareness

This code was referenced 19 times and was the most frequently used code under this theme, suggesting that the participants believe messaging about PrEP, centering heterosexual women, is an important factor that must be addressed if we want to increase PrEP initiation among cis Black women.

During focus group 6, participant 1 made references to the messaging that she has seen for the HPV vaccine when she said,


*‘Just like that doggone, HPV vaccine and Gardasil. I want all of that. I want the same energy. I want women on all types of media, like the commercials. Thank you for showing Black men and trans women and I want to see some regular women in there. I want to see different types of relationships. I want to see young and old people because the senior citizens, they wilding out there too. I want it to be like, it’s kind of like diversity and inclusion, but I want the real one, not the watered down one that y’all keep giving us in corporate America. Give me the actual diversity and inclusion everywhere. Just like, yeah, just an apple a day keeps a doctor away. I want to know about PrEP if I’m sexually active.’*


This sentiment from participant 1 illustrates her desire for diverse and inclusive messaging around PrEP where people from all walks of life are represented, including representation of people who look like her.

2.Code: Willingness to take PrEP

Willingness to take PrEP is a code that applies to varied themes; however, in the context of this theme, willingness to take PrEP is a necessary precursor to PrEP uptake and must be present to facilitate PrEP initiation.

In response to the facilitator asking whether PrEP as an injectable makes a difference in their willingness to take it during focus group 5, participant 3 stated,


*‘In my personal opinion, I don’t mind taking a pill every day because I have a reminder on my phone. So, me personally, because I have a very, very busy schedule, you would have to take time off of work to go and get the injectable. Don’t get me wrong, it may work for some women who can’t remember to take the pill every day, but for some of us that has a very busy schedule. If I have the pill right there in my area, I’ll just take it versus whether to take it off of work and going to go get the injectable. But maybe it’ll be easier for some women who don’t want to take the pill every day.’*


Participant 3 describes how her willingness to take PrEP is filtered through the modality because of the restraints her busy schedule has on the feasibility of being adherent to the medication.

##### Theme 6: Factors That Inform the Choice for or against PrEP Use among Cis Black Women

This theme brings together a diversity of expressed ideals whereby participants discussed reasons why, or why not, cis Black women might take PrEP. In order to gather a balanced perspective of the factors that inform the decision for PrEP uptake, this theme aims to present a well-rounded picture.

Code: Reasons why cis Black women might be willing to take PrEP

This quote was used six times and describes some of the reasons why cis Black women may choose to take PrEP. These reasons inform willingness and share insights into the decision process for PrEP initiation.

During focus group 5, participant 3 describes her belief regarding changes in the perspective of cis Black women as it relates to willingness to take PrEP when she responds to the facilitator who asked about their thoughts on why women would or would not choose to use PrEP when she said,


*‘I believe that women would choose, especially Black women to use PrEP because I feel like we’re definitely becoming more, what’s the word, sensitive or aware of the things that we need in order to stay healthy, in order to be around.’*


Participant 3 is describing an increase in perceived risk of HIV among cis Black women, which she believes is supporting their willingness to use PrEP for HIV prevention.

2.Code: Scientific data are not always reflective of cis Black women

Although this code was used only four times, exposing the awareness that cis Black women have regarding their absence and lack of representation in research is worth noting. The women described how failure to center their voices and lived experience in research compounds the distrust and medical mistrust that already exists; however, their desire to participate and engage in the research process is fully expressed here.

During focus group 5, the facilitator led a conversation about trust in the healthcare system. Participant 1 stated,


*‘I think if they did more researches like this for Black women, I think it would definitely build some more trust because we don’t have enough information. And a lot of times when they do focus groups, it’s not really with Black women, it’s a lot on the Caucasian and male or female, because I used to work in the medical center and most of the time they would be geared towards other races. Not saying that they’re against Blacks, but they just don’t promote it towards Blacks. So, I don’t know how they can build that to where Blacks and Black women would want to join. But that would be great if we had more resources as far as information that was put into focus groups or research.’*


Participant 1 called for more research that prioritizes the health of cis Black women. She describes that offering more education and research opportunities may foster trust.

##### Theme 7: Willingness for Self-Advocacy with Healthcare Provider

The codes here describe the willingness of participants to engage their healthcare providers in shared decision-making about their sexual health. Under this theme, willingness to take PrEP, advocate for oneself with the provider, and educate others about PrEP are explored.

Code: Willingness to take PrEP

The code willingness to take PrEP is applicable to this theme in a way that situates the participant’s willingness as secondary to witnessing the lived experience of other people with health conditions that could have been prevented. Although the initial motivation here was external, the commitment to self-advocate for one’s personal health became internal and normalized as a new way of engaging in prevention options such as PrEP.

During focus group 2, the facilitator led a conversation about self-love and self-respect. Participant 1 spoke about her extended family and her desire to stay healthy so that she can be here for them. She mentioned some previous occurrences that threatened her health. She then stated,


*‘So, anything that I can help to help me prevent, yes. And like I said, like I explained to y’all earlier, my mother is a HIV patient, so if I can help myself prevent, yes.’*


Through the lens of self-advocacy, participant 1 is motivated to protect her health because of her extended family, which can fuel the behavior needed to self-advocate with a healthcare provider. She also has access to vicarious learning through her mother who has HIV, a diagnosis that she wants to prevent for herself.

2.Code: Willingness to advocate for self with healthcare provider

The quotes with this code describe the rationale behind the willingness to move beyond personal discomfort and commit to advocating for oneself with healthcare providers.

During focus group 6, the facilitator asked if participants were comfortable having conversations with their physicians about their sexual health. In response, participant 1 stated,


*‘Now I am 29, the answer is yes. Now, when I was younger, especially in college when I didn’t have insurance and I was going to the free clinic and things like that, I wasn’t very, and I was in Alabama, so I didn’t really feel like I could ask those questions. But now that I sought out a Black OB-GYN, which unfortunately all skin folk aren’t your kin folk, I’ve had to be a little bit more vocal of just because I am a single Black woman, that doesn’t mean that my reproductive health or my sexual health are still not priority. So, do I feel comfortable enough to say something? Yes. Whether or not she listens to me or not, that’s another story. But yeah, I feel comfortable enough pressing for my own, being my own advocate now. But that’s like recent.’*


This response offers the reader a great deal of data on how the participant perceives age, location, insurance status, race, and gender influenced her ability to engage in shared decision-making with her healthcare provider. We have witnessed a growing confidence in recent years.

##### Theme 8: Acceptance of PrEP as Approved

This theme explores the perspectives of participants in accepting PrEP in its approved modalities to date, as a pill and an injection, with the side effects and associated costs.

Code: Acceptance of side effects with PrEP

This code explores how participants described an understanding that side effects are a part of medication uptake; however, the potential health benefits of the medication often make the benefits outweigh the risks.

During focus group 5, the facilitator asked participants about their thoughts on the possible side effects of taking PrEP. Participant 3 said,


*‘So, in terms of side effects, I’ve pretty much gotten past the thought of just being hyper-focused on that, because I think COVID definitely did it for me where it was just like, okay, you got to take this vaccination if you want to avoid transmitting or receiving COVID. So from that standpoint, I’m more comfortable.’*


Participant 3 expresses a former hyper-awareness of the side effects and a change in philosophy post-COVID-19 when she witnessed the positive health implications of the COVID-19 vaccine at preventing viral transmission. This increased her perceived value of PrEP uptake relative to the side effects.

2.Code: Acceptance of longevity of PrEP regimen

This code was used only once and the quoted text below describes what the participant conveyed, which is that long-term health preservation is aligned with long-term medication uptake.

Participant 4 described her thoughts on whether the side effects of PrEP uptake outweighed the benefits during focus group 5. She said,


*‘But what I will say is it’s taking something long term that it was like, okay, it’s like a vitamin that I’ll have to add to my regimen and just know that it’s something that I have to continue. So, if that ever changed, I’d be like, this is great. But I do know that that’s one of the things.’*


She compared PrEP to a vitamin with properties to preserve long-term health. Even though vitamins and PrEP are preventive, she describes them as a necessary behavior.

##### Theme 9: Awareness of HIV Risk with Drug Use

Although this study was not centered around substance use, substance use did become a part of the conversation and was captured in this theme, whereby some participants shared an awareness of the link between risks associated with HIV and substance use, both generally and personally.

Code: Awareness of HIV risk due to personal drug use

There was one use of this code where the participant described her lived experience with substance use in the past and how over time, her behavior has changed. She now adopts a more proactive way of engaging in HIV prevention with recognition of her historic vulnerability to HIV.

During focus group 3, the facilitator provided feedback regarding the honest portrayal of a former addiction by participant 1. The conversation was about perceived monogamy versus actual polygamy in heterosexual relationships. The facilitator stated that participant 3 was quiet when she asked her to join the conversation. Participant 3 stated,


*‘I was vulnerable at a time. I’m an ex-drug user, so I was out there doing bad things. But now that I’m wiser, I think it’s good to get tested regularly.’*


Participant 3 shared vulnerability and highlighted her awareness that her former drug use placed her at significant risk for HIV. With more awareness, her behavior has changed and she is now engaged in routine HIV testing.

##### Theme 10: Perceived HIV Knowledge

The final theme explored the quoted text that reflected participants’ thoughts and beliefs about their HIV knowledge relative to how HIV is transmitted.

Code: Awareness of HIV transmission routes

This code was used once when the participant described blood transfusions as a transmission route for HIV.

During focus group 2, the facilitator asked participants if they thought that PrEP is for them based on what they learned from commercials. She then asked them who they thought PrEP was for. Participant 1 stated,


*‘My mother diagnosed with HIV. My mother’s still living, thank God. I can’t remember the damn year, though, because it was so long ago. And the first thing I thought about is, “I’m fixin’ to lose my mother.” My mother got it through a blood transfusion. But she’s been good. Mother’s been good. She takes medicine or whatever and then she go sees a doctor and she let us know. She lets us know how her T-cell count and stuff is like that. I mean, so it is in my family, But I would’ve never thought… All I thought about was gay men when I did see it. Yeah.’*


Participant 1’s response demonstrates her perceived HIV knowledge regarding transmission routes in the case of her mother, who she is sure contracted HIV through a blood transfusion. She also demonstrates knowledge of her mother’s clinical visits and metrics for assessing immunity.

## 4. Discussion

The findings of this study provide a roadmap to design interventions that may allay concerns about PrEP and can empower cis Black women with the tools and information needed to actively engage healthcare providers in shared decision-making about their sexual health. Although participant responses inform the scientific community of diverse reasons why cis Black women do not/have not engaged readily with PrEP, the presentation of those diverse reasons was organized in this paper by factors that went beyond their control (i.e., an external locus of control, requiring buy-in and action from healthcare providers and/or community) versus factors that were within the control of cis Black women (i.e., an internal locus of control). Studies assessing locus of control with women are limited; thus, this study’s findings increase the available literature on this topic relative to the decision-making power of women in general. The rationale for framing the findings in this way is to better describe and understand the direction of pathways to motivate PrEP uptake among cis Black women at the individual and/or social and structural levels.

### 4.1. External Locus of Control

For individuals with an external locus of control, their decisions are largely governed by the power of coercion [22]. Specifically, these individuals believe that their lives are out of their control and are governed by chance or by external forces that may include other people whom they perceive as more powerful [22,25,26,27,28,33,34]. As such, these individuals use the power of persuasion and expertise [22] to influence decisions for themselves and others. This way of being is also associated with a feeling of powerlessness, as the external environment seemingly has power over the lived experience of these individuals. Lack of awareness and knowledge of PrEP [35,36], medical mistrust of healthcare providers [37,38,39,40,41], and social stigma linked to PrEP use [40,41,42,43,44] are commonly described as barriers in the external environment to PrEP use among cis Black women [45]. Efforts to overcome these external factors require facilitators of the decision to adopt PrEP for HIV prevention among cis Black women. Participants in this study described preferences for healthcare providers with racial and gender-based concordance. When evaluated through the lens of an external locus of control, similarities in social identifiers may enhance comfort for cis Black women when they have the responsibility of self-advocacy with healthcare providers about their own health. Codes linked to an external locus of control revealed that some participants were influenced by media messaging on PrEP, particularly television-based media messaging on PrEP and the experience of not seeing themselves represented in media [46,47]. Based on this external visual, several participants self-excluded themselves as PrEP-eligible. These findings align with the recent literature stating that ‘Black women do not see themselves and their community in PrEP commercials and advertisements’ [12,46]. Furthermore, the primary focus of PrEP-focused media messages on men who have sex with men is seemingly unrelatable to cis Black women [46]. The narrative shaped by the lens of an external locus of control suggests that some cis Black women are influenced by external factors, which can serve as a barrier or facilitator to their decision to take PrEP for HIV prevention.

### 4.2. Potential Pathway from External Locus of Control to Internal Locus of Control

Based on the summation of the qualitative data presented, a preference for certain provider types is connected to self-advocacy because the comfort with that provider creates the space for self-advocacy. It seems as though when cis Black female participants felt comfortable communicating, they felt comfortable being honest, speaking their mind, and engaging in a conversation as their full authentic selves, which seemed to empower self-advocacy. The external locus of control here is the healthcare provider who either possesses those qualities or creates that environment. When those elements are absent in the external environment, it appeared as though the external environment failed to catalyze the internal locus of control to spark self-advocacy, a quality that is harnessed and controlled by the internal locus of control.

### 4.3. Internal Locus of Control

Individuals with an internal locus of control lean into their self-confidence to shape their decision-making. They believe that they have the internal power to influence outcomes in their life [22,25,26,27,28,33,34]. This approach to life influences the way these individuals make health decisions and the variables used and needed to assist in the decision-making process. Literary findings suggest that among cis Black women, an inverse relationship exists between the awareness of a lack of monogamy from sexual partners and the belief that PrEP offers protection, self-control, and empowerment [46,48,49]. The current study produced findings illustrating that codes associated with an internal locus of control were more often related to personal agency in determining PrEP eligibility, clarity about the safety and empowerment that PrEP offers, a higher perceived risk of HIV within sexual relationships, and a greater willingness to self-advocate with a healthcare provider and consider PrEP as an option. There seems to be an alignment between heightened self-awareness and confidence in one’s knowledge and ability to make correlations between perceived risk and willingness to take PrEP. This means that an individual who operates with an internal locus of control, likely reinforced by their life experiences and built environment, may be more likely to choose PrEP as an option compared to those with an external locus of control. This hypothesis aligns with literary findings stating that people with an internal locus of control invest more in prevention and health capital [22]. The adoption of PrEP as an HIV prevention method among individuals with an internal locus of control seems to be aligned with a personal investment in one’s personal sexual health.

### 4.4. Interplay of Locus of Control and Determinants of Health

An external locus of control can be fueled by structural, socioeconomic, and sociocultural factors that constrain an individual’s ability to make health decisions. For individuals with significant vulnerability to HIV, these factors sometimes cannot be overcome [50]. Due to the inherent limitations of an external locus of control on the preservation of an HIV-negative status amidst myriad determinants of health, primary prevention strategies for HIV are more poised for short-term success if the outcome is focused on individual-level behaviors instead of expectations to overcome the external environment [51]. Instead, the establishment of new pathways to behavioral change should include the adoption of ‘ways of being’ that are innate to individuals with an internal locus of control. Equipping HIV-vulnerable populations to engage healthcare providers in sexual health discussions, enhancing their personal knowledge of PrEP and its potential use in their sexual health practices, and promoting a healthy balance of responsible sex that includes routine condom use to prevent sexually transmitted infections (STIs) have the potential to motivate an increase in PrEP use among cis Black women. It is important for interventionists to manage empowerment potential while preventing the development of a sense of invincibility on PrEP among participants [46]. Determinants of health do impede full autonomy of overcoming HIV vulnerability among cis Black women. However, avenues to expand the utility of a locus of control to maximize personal power in the preservation of sexual health is a plausible pathway worth pursuing among interventionists and researchers engaged in the PrEP care cascade.

### 4.5. Limitations

Our findings must be interpreted within the context of some study limitations. Although the study team had multiple coding sessions, both virtual and in-person, the final assessment with the Kappa statistic failed to demonstrate agreement between coders, limiting the generalizability of the findings. The research experience level of focus group facilitators varied between coders. Similar to other qualitative studies exploring facilitators and barriers to PrEP initiation among Black women [6,52], the inclusion of small samples limits the generalizability of the study findings to referent other cis Black women in Texas and in the South, where HIV vulnerability is highest for cis Black women. Questions about sexual behaviors may be of a sensitive nature, which could lead to social desirability bias. It is possible that some participants may have understated their focus group engagement in part due to concerns of stigma around their sexual behaviors to align with expectations of social norms. The study utilized a single focus group facilitator across all seven focus groups. In each focus group discussion, the facilitator established a welcoming and non-judgmental atmosphere, validated all participants’ contributions to the conversation, and encouraged anonymity and confidentiality by allowing participants to keep their videos off if desired, requiring they change their screen name to a de-identified participant identification (ID) number (i.e., 001), and referring to participants by their participant ID number throughout the session. This qualitative research serves a vital role in exploring the framing of the lived experiences of cis Black women through an internal or external locus of control to discern the importance of facilitators and barriers to PrEP adoption. Enhancing the volume of the available literature focused on promoting PrEP initiation with cis Black women is a priority.

## 5. Conclusions

In conclusion, evidence of a persistent and disproportionate burden of HIV among cis Black women in EHE jurisdictions, including Houston, TX, highlight the need for interventions that can effectively challenge the byproducts of determinants of health and empower increased capacity to leverage internal power to influence personal and community sexual health outcomes. By organizing PrEP facilitators and barriers through the dichotomy of an internal and external locus of control, we contribute to the literature a different way of exploring intervention strategies with cis Black women in order to move the needle on increasing PrEP uptake with this HIV-vulnerable population, in alignment with the EHE initiative through the HIV prevent pillar. The prevent pillar focuses on preventing new HIV transmissions by using proven interventions, including PrEP and syringe service programs. Despite varied intervention approaches that have failed to increase PrEP initiation among cis Black women, in tandem with scientific evidence of this biomedical intervention’s efficacy at preventing HIV transmission when taken as prescribed, the opportunity to identify new approaches to connect with the sexual health decision-making practice of this population exists. The findings of this paper and framing with a locus of control can help interventionists and researchers to assess their population differently and tailor intervention approaches to the ‘way of being’ of each client based on their locus of control. This approach has the potential to connect more securely with the natural ‘way of being’ of participants, making it easier to integrate intervention strategies into their daily lives.

## Figures and Tables

**Table 1 ijerph-21-01084-t001:** Descriptive data describing participants who completed the screening survey.

Variables	Subcategories	Respondents (241)	Study Participants	Respondents	Study Participants
Biological Sex		N = 230	N = 20	%	%
	Female	226	20	98.26%	100.00%
	Male	4	0	1.74%	0.00%
	Unsure	0	0	0.00%	0.00%
	Missing	0	0	0.00%	0.00%
Age		N = 225			
	<18 years	1	4	0.44%	20.00%
	18–29 years	148	4	65.78%	20.00%
	30–39 years	49	4	21.78%	20.00%
	40–49 years	17	7	7.56%	35.00%
	>49 years	10	5	4.44%	25.00%
Race		N = 164			
	African American	82	13	50.00%	65.00%
	Black	81	7	49.39%	35.00%
	Other	1	0	0.61%	0.00%
City of Residence		N = 142			
	Houston, TX	140	19	98.59%	95.00%
	Other	2	1	1.41%	5.00%
English Speaking		N = 213			
	Yes	193	20	90.61%	100.00%
	No	20	0	9.39%	0.00%
Sexual activity with a male partner within the last 12 months		N = 224			
	Yes	214	20	95.54%	100.00%
	No	10	0	4.46%	0.00%
Access to a phone and/or internet		N = 191			
	Yes	189	20	98.95%	100.00%
	No	2	0	1.05%	0.00%
Ability to provide informed consent		N = 191			
	Yes	180	20	94.24%	100.00%
	No	11	0	5.76%	0.00%
Current HIV Status		N = 180			
	HIV-positive	16	0	8.89%	0.00%
	HIV-negative	164	20	91.11%	100.00%
Willing to participate		N = 160			
	Yes	158	20	98.75%	100.00%
	No	2	0	1.25%	0.00%

**Table 2 ijerph-21-01084-t002:** Themes and associated codes identified through thematic analysis of focus group data.

Category of Themes	Themes	Codes (*n* = 1217)	Frequency of Codes
*External Locus of Control*			
	Self-advocacy with providers about health	*7 codes*	
		Preference in having a Black provider	14
		Preference in having a female provider	10
		Provider not discussing PrEP	5
		Provider not discussing sexual health	5
		Relationship with provider supports comfort to discuss PrEP	3
		Provider comfort talking about sexual health	2
		Perception that provider’s knowledge is outdated	1
	Media Influences on understanding of PrEP	*6 codes*	
		TV media messaging about PrEP	60 **
		Perception that cis Black/African American women are not prioritized in TV media messaging about PrEP	26 *
		Media messaging suggests that PrEP is for LGBTQ+ community	19
		Females may feel excluded due to lack of representation in media marketing	7
		Perception that PrEP commercials were not relatable due to lack of representation	2
		Lack of PrEP awareness due to disconnect from media	1
	Comfort with provider–patient interactions	*12 codes*	
		Perceived comfort with discussing sexual health with provider	8
		Communicating openly with your provider is part of your health	8
		Perceived comfort in having a female provider	5
		Comfort talking about PrEP with provider	5
		Perceived trust in provider	4
		Discomfort communicating with provider about sexual health	4
		Long-term relationship with provider promotes comfort to discuss sexual health	2
		Perceived comfort in having a Black provider	2
		Honest communication as a remedy for medical mistrust	2
		Comfort with talking about PrEP with provider regardless of provider’s gender	2
		Perceived higher level of trust in having a Black provider	1
		Acknowledged the need for transparency in health communication with providers	1
	Provider engagement in sexual health of cis Black female patients	*10 codes*	
		Experience with provider offering HIV testing	13
		Willingness to engage provider in HIV testing	10
		Experience with provider discussing sexual health	5
		Level of trust influencing decision to get HIV test	4
		Experience with provider offering STD testing	3
		Patient desires to have preventative conversations with provider	2
		Hesitancy for conversations about PrEP due to perceived stigma or judgment	2
		Assumption of sexual risk for provider	2
		Level of trust within relationship influencing decision to take PrEP	1
		Experience with provider discussing PrEP	1
	Awareness of peers with HIV and/or taking PrEP	*4 codes*	
		Awareness of peers on PrEP	2
		Awareness of peers with HIV prompts willingness to learn more	2
		Awareness of peers taking PrEP opens communication about PrEP	1
		Awareness of peers with HIV and PrEP experience	1
	PrEP-related concerns	*5 codes*	
		Concern with cost and insurance coverage for PrEP	4
		Concerns for PrEP injectable	4
		Cost as a barrier to PrEP readiness	3
		Concerns for PrEP as an injection	2
		Concerns for PrEP oral pill	2

Legend: ** 50 or more uses of the code; * 20–49 uses of the code.

**Table 3 ijerph-21-01084-t003:** Associated codes of the theme Internal Locus of Control.

Category of Themes	Themes	Codes (*n* = 1217)	Frequency of Codes
*Internal Locus of Control*			
	Perceived PrEP knowledge	*13 codes*	
		Perception that PrEP is not for heterosexual people	28 *
		Perception that PrEP is for gay men	24 *
		Perception that PrEP eligibility is based on sexual orientation	23 *
		Concern about PrEP’s side effects	19
		Perception that PrEP is for people who inject drugs	13
		Perception that PrEP is for people who inject drugs	13
		Research participation serving as access to PrEP knowledge	12
		Perception that PrEP is for risky lifestyles	11
		Concern about PrEP’s side effects with pre-existing conditions	9
		Perception that taking PrEP promotes risky behavior	4
		Personal knowledge that PrEP is for women	3
		Concern for PrEP not being studied in cis women	1
		Awareness of lack of knowledge on PrEP may prompt willingness to learn more	4
	PrEP Willingness	*6 codes*	
		Stigma around PrEP uptake	10
		Reasons women might be hesitant towards taking PrEP	8
		Reasons why cis Black women might be willing to take PrEP	6
		Awareness of lack of knowledge on PrEP may prompt willingness to learn more	4
		Self-awareness of lack of knowledge on PrEP	2
		Certainty that PrEP is not an option	2
	Knowledge of PrEP as a prevention tool	*8 codes*	
		Perception that PrEP offers safety	29 *
		PrEP empowers ability to control personal sexual health	20 *
		Perception that PrEP offers freedom	19
		Knowledge of PrEP as preventing HIV	17
		Research participation serving as access to PrEP knowledge	12
		Personal experience with taking PrEP	3
		Understanding that PrEP can prevent HIV risk to others	3
		Knowledge that PrEP helps with HIV transmission	2
	Awareness of sexual risk	*11 codes*	
		Awareness of HIV risk within sexual relationships	28 *
		Perception of elevated HIV risk with new partner	8
		Personal investment in knowing one’s personal HIV status	6
		Perception that PrEP is for risky lifestyles	11
		Awareness of HIV risk	11
		Perception that taking PrEP promotes risky behavior	4
		Awareness of changes in sexual risk over time	4
		Awareness of STI risk within sexual relationships	2
		Perception of elevated HIV risk with multiple partners	1
		Acceptance of risk for getting STDs	1
	Perceived facilitators of the decision to use PrEP	*6 codes*	
		Perception that varied PrEP messaging can bring awareness	19
		Willingness to take PrEP	15
		Perception that awareness and education of PrEP prompts willingness to take PrEP	8
		Perception that PrEP will not impact their sexual intimacy	8
		Willingness to take PrEP as an injection	4
		Perception that PrEP can reduce personal risk of getting HIV	2
	Factors that inform the choice for or against PrEP use among cis Black women	*2 codes*	
		Reasons why cis Black women might be willing to take PrEP	6
		Scientific data are not always reflective of cis Black women	4
	Willingness for self-advocacy with healthcare provider	*2 codes*	
		Willingness to take PrEP	15
		Willingness to advocate for self with healthcare provider	4
		Willingness to educate others on PrEP	1
	Acceptance of PrEP as approved	*2 codes*	
		Acceptance of side effects with PrEP	3
		Acceptance of longevity of PrEP regimen	1
	Awareness of HIV risk with drug use	*2 codes*	
		Awareness of HIV risk due to personal drug use	1
		Awareness of personal HIV risk due to previous drug use	1
	Perceived HIV knowledge	*1 code*	
		Awareness of HIV transmission routes	1

Legend: * 20–49 uses of the code.

## Data Availability

The data presented in this study are not available because the data are qualitative and the words, the situations presented, and phrasing of the wording can potentially identify participants and compromise confidentiality.

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
