# Peer review of "An Examination of Perceptions among Black Women on Their Awareness of and Access to Pre-Exposure Prophylaxis (PrEP)"

_ijerph, 2024, doi:10.3390/ijerph21081084_

Round 1

Reviewer 1 Report

Comments and Suggestions for Authors

Overall, a well-written article and covers a very important and often overlooked topic. I appreciate the efforts to better understand how to increase PrEP access among cis Black women – this article helps fill a large gap in the literature and gives real direction in how to increase internal locus of control among cis Black women to increase PrEP uptake. A few recommendations to further improve the manuscript:

End the HIV Epidemic (EHE) is mentioned a few times throughout the manuscript but not defined – it should be defined and referenced when first introduced. In the conclusion, a HIV prevent pillar is mentioned, but not defined – this would benefit from a sentence or two providing context.

The introduction notes social and structural discrimination and biases that keep cis black women oppressed – a sentence or two elaborating what these social and structural discriminations are, and how they keep cis black women oppressed, should be cited and would provide helpful context.

The introduction notes that less than 2% of eligible cis black women take PrEP – how is eligible defined, for whom is PrEP being recommended?

The end of the second paragraph in the introduction notes that no significant decreases in HIV or increases in PrEP has been observed – this needs a citation.

The third paragraph in the introduction starts by highlighting a need to bridge the gap between healthcare providers extending access to PrEP and eligible cis black women being ready to take PrEP – this would benefit from a sentence or two defining the gap – are healthcare providers not extending access to PrEP, are women declining PrEP when offered, etc.

The last sentence of the third paragraph is a bit long and difficult to follow – consider breaking into two sentences.

There were a few instances where Black women was used vs. cis Black women – remain consistent throughout.

In the fourth paragraph of the intro (line 63), need for increased education is listed, but it seems like receipt of education may be what the author meant.

The end of the fourth paragraph of the intro notes that PrEP initiation is an important and necessary behavior change – consider for whom among cis black women this is true, i.e. those vulnerable to HIV.

Line 96 – the word “at” seems to be a typo

Define study eligibility.

In the third paragraph in the Materials and Methods section, the FGD and base-line survey are mentioned – are these the same thing?

The third paragraph in the materials and methods section also mentions Theory of Gender and Power and The Sexual Script – There should be a few cited sentences describing what these are.

It seems like the focus groups were online as it was noted videoconferencing software was used – the manuscript should explicitly detail how the groups were conducted, in person or virtual. The limitations section also noted a variance between sites – more description of the sites (all different facilitators? Research experience? Number of participants in each) would be helpful. The variance between sites should either be mentioned in the results and/or clarified/defined in the limitations – as it is now, it is not clear to what the variance is referring.

Line 119 also mentions the next phase of the study for the first time – It would be helpful if the next phase was introduced at the end of the introduction, where it describes the current study as a necessary first step. Reference at line 119 then would have appropriate context.

Lines 121 and 122 mention the theory of planned behavior – a citation is needed here.

Line 128 looks to have a type – live instead of life

Line 134 and 135 note specific counties – consider describing location rather than giving specific county names (rural, urban, southern, etc.)

Table 1 describes the participants who filled out the screening survey, yet only 20 participants were included in the focus groups. How did the team get from the 241 responses to only 20 participants? Consider providing demographics of the participants, possibly in comparison to the 241 responses to consider whether the 20 participants are representative of the original 241 responses.

Table 1 has only two choices for race – African American or Black – were any other choices provided? Also, there are different N for each variable – why? Did some participants only fill out some variables?

The sentence spanning lines 149 – 151 is unclear.

In the Results, it is noted that there are 1,217 codes; how many are among the highest frequency/listed in the table/included in descriptions vs. how many are left out? Also, how was it determined which codes would be described – there are multiple instances where codes with only 1 observation are described in the text and listed in the table.

Consider revising formatting to remove bullets for the themes – continuation of numbering system would be easier to follow. Line 180, 3.11 External Locus of control, should be 3.1.1. Theme 1 in that section could then follow 3.1.1.1.

The overall presentation of the Results would benefit from more summary – Themes are provided, but with no description. A few sentences describing what responses had in common in each theme, why reviewers grouped codes/responses in the given theme, etc., would be helpful. Likewise, a summary of each code would be helpful rather than only the single highlighted quote, especially in cases where 20+ responses are included under 1 code. For example, lines 207 – 215 describe 1 example of media influence, but there are 60 uses of this code – a summary of the different points covered in those 60 uses, and possibly another direct quote or two, would be helpful. The quotes selected and analyses thereof were well done and easy to follow. Noting how many responses are covered in each code, and a summary of the different points covered, would provide more context for the results.

Throughout description of the results, it is also noted frequently that the facilitator of a specific group asked a specific question – this gives the impression that not all groups received the same questions. Clarification is needed, and if all groups did not receive the same questions, that should be addressed.

Lines 306 – 311 describe cost related to PrEP – the quote reads like it is referring to personal cost, not financial, though is coded as financial cost – is there more context to the quote that led it to being classified as financial cost rather than a personal cost?

Consistency should be used when referencing groups and participants – manuscript changes from numerals (ex. 3) to words (ex. Three).

Line 393 is Reasons women might be hesitant towards taking PrEP – this seems like a different format than other codes, and the example quote provided looks like it could fit with other previously provided codes. Also, the title of the code implies that multiple disconnected responses could be grouped under this code. In the analysis of the quote, it notes there is a hesitancy in taking medication that is not needed, and 57% of the responses referred to vaccine hesitancy – it seems like these statements would better inform the name of the code. Additionally, noting that 57% of the responses referred to vaccine hesitancy is very helpful – statements like this would help the reader better understand the constellation of responses – this sort of summary is what is meant by summaries of themes/codes.

Line 426 notes how quotes from two different participants coded in two different codes are similar – again, summaries of the codes would help the reader understand how they differ rather than leaving them wondering why they aren’t grouped together under the same code.

Line 551 has a possible type – addition rather than addiction.

Line 561 notes 2 code as the first code under theme 10

Table 3 – perception that PrEP is for people who inject drugs is in twice

In the discussion, internal vs. external loci of control is the focus – did participants exhibit responses that indicated both internal vs. external loci of control, or did individuals skew either more internal vs. external? Assuming they most often displayed a mix of both, highlighting the need to address the internal locus of control via intervention makes sense; however, if a significant number of participants expressed only external locus of control in their response, such an intervention runs the risk of missing the needs of those who only exhibit an external locus of control.

The sentence in lines 656 through 659 needs citation.

The last paragraph in the limitations is only one sentence – it should either be combined to the previous paragraph or have additional statements added.

Author Response

Comment: End the HIV Epidemic (EHE) is mentioned a few times throughout the manuscript but not defined – it should be defined and referenced when first introduced.

Response: We appreciate this recommendation. We have defined ‘End the HIV Epidemic’ when it is first referenced and introduced. We also added a citation.

Comment: In the conclusion, a HIV prevent pillar is mentioned, but not defined – this would benefit from a sentence or two providing context.

Response: In the introduction section, we added a description of the ‘Ending the HIV Epidemic’ initiative, which included mention of ‘prevent’ which is one of four key strategies or pillars outlined in the ‘Ending the HIV Epidemic’ initiative with a corresponding citation in paragraph 1 of the introduction section on lines 30-33, page 1. In the conclusion paragraph 1, page 27, lines 956-958, we provided additional context as to what is meant by the ‘HIV prevent pillar’. 

Comment: The introduction notes social and structural discrimination and biases that keep cis black women oppressed – a sentence or two elaborating what these social and structural discriminations are, and how they keep cis black women oppressed, should be cited and would provide helpful context.

Response: We appreciate this suggestion. We have added two sentences with a corresponding citation that contextualizes the social and structural discriminations that impede HIV prevention efforts for cis Black women on page 1, paragraph 1 of the introduction section, lines 39-42.

Comment: The introduction notes that less than 2% of eligible cis black women take PrEP – how is eligible defined, for whom is PrEP being recommended?

Response: In Conley et al. (2022) paper, they cite PrEP-eligibility criteria as described in Kulie et al. (2020) paper, which included an expansion of 2020 CDC’ PrEP eligibility criteria. A direct quote from the article states, ‘We estimated the number of additional patients who would be PrEP eligible using the following criteria: (1) positive result from an STI test conducted during the index ED visit for chlamydia, gonorrhea, or syphilis and not in a mutually monogamous sexual partnership; (2) positive result from an STI test conducted during the index ED visit for chlamydia, gonorrhea, or syphilis, BV or trichomoniasis and not in a mutually monogamous sexual partnership; (3) self-reported STI in survey or positive result from an STI test conducted during the ED visit (for any of the five aforementioned STIs) regardless of monogamy; and (4) assuming that sexual partners of participants who stated that they did not know or refused to indicate the HIV status of their partner were HIV infected.’

Current CDC recommendations for PrEP-eligibility includes all people who request access to PrEP.

References:

Conley C, Johnson R, Bond K, Brem S, Salas J, Randolph S. US Black cisgender women and pre-exposure prophylaxis for human immunodeficiency virus prevention: A scoping review. Womens Health (Lond). 2022 Jan-Dec;18:17455057221103098. doi: 10.1177/17455057221103098. PMID: 35699104; PMCID: PMC9201306.

Kulie P, Castel AD, Zheng Z, Powell NN, Srivastava A, Chandar S, McCarthy ML. Targeted Screening for HIV Pre-Exposure Prophylaxis Eligibility in Two Emergency Departments in Washington, DC. AIDS Patient Care STDS. 2020 Dec;34(12):516-522. doi: 10.1089/apc.2020.0228. PMID: 33296271; PMCID: PMC7869869.

Comment: The end of the second paragraph in the introduction notes that no significant decreases in HIV or increases in PrEP has been observed – this needs a citation.

Response: We have added a citation to support this statement in paragraph 2 of the introduction section, page 2, line 54.

Comment: The third paragraph in the introduction starts by highlighting a need to bridge the gap between healthcare providers extending access to PrEP and eligible cis black women being ready to take PrEP – this would benefit from a sentence or two defining the gap – are healthcare providers not extending access to PrEP, are women declining PrEP when offered, etc.

Response: We have added a few sentences on page 2, paragraph 3 of the introduction section, lines 57-62, defining the gap. This section now reads as, ‘Bridging the gap between healthcare providers extending access to PrEP during clinical visits and eligible cis Black women being ready to take PrEP requires a growing awareness of why the gap exists in the first place. Survey data demonstrate that some healthcare providers in Texas are not comfortable prescribing PrEP.6 A single-arm quantitative pilot study among women in Texas demonstrated interest in and willingness to use PrEP, but they were not current PrEP users.7 Thus, when women are supported to access PrEP, it is imperative that they are received well by knowledgeable HCPs who can and will offer PrEP. With respect to healthcare providers,…

Comment: The last sentence of the third paragraph is a bit long and difficult to follow – consider breaking into two sentences.

Response: We agree and have broken up the last sentence of the third paragraph into three sentences. It now reads, on page 2, paragraph 3 of the introduction section, lines 64-70, as, ‘Achieving equitable access to PrEP requires consideration of the healthcare system and environment. Specifically, access to a discreet and convenient clinic, insurance coverage, and availability of comprehensive services in specified locations to assist clients with PrEP access. Lastly, receiving navigation services, in addition to the aforementioned services, were perceived as a healthcare benefit (64% of clients receiving services said the initiated PrEP) and were all listed as facilitators of PrEP initiation.5

Comment: There were a few instances where Black women was used vs. cis Black women – remain consistent throughout.

Response: We reviewed the use of ‘Black women’ and ‘cis Black women’ throughout the manuscript. We have remained consistent with the use of ‘cis Black women’ except in instances in which statements are derived from the literature where the distinction was not made and merely refers to ‘Black women’ as a whole.

Comment: In the fourth paragraph of the intro (line 63), need for increased education is listed, but it seems like receipt of education may be what the author meant.

Response: We changed “need for increased education” to “provision of education” on page 2, paragraph 4, line 76.

Comment: The end of the fourth paragraph of the intro notes that PrEP initiation is an important and necessary behavior change – consider for whom among cis black women this is true, i.e. those vulnerable to HIV.

Response: The authorship tip team appreciates this comment and thoughtful consideration. Given the change in CDC guidelines regarding PrEP eligibility whereby eligibility is no longer limited to a risk-based criteria and is far more global, deeming anyone who expresses an interest in PrEP eligible for this preventive biomedical intervention, our authorship team reaffirms that the original message that PrEP initiation is an important and necessary behavior change. Furthermore, given the disproportionate burden of HIV vulnerability to all cis Black women, inclusive of diverse socioeconomic status, education level, insurance status, and varied sociodemographics that contribute diversity within this cohort of women, vulnerability to HIV is a matter that affects cis Black women globally, aligning with updated CDC recommendations.

Comment: Line 96 – the word “at” seems to be a typo

Response: Thank you for pointing this out. We have removed the word, “at” on page 3, paragraph 2 of the Materials and Methods section, line 117, as it was a typo.

Comment: Define study eligibility.

Response: Study eligible was added in Section 2. Materials and Methods, paragraph 2, lines 118-124. It now reads, ‘We recruited a purposive sample of 20 PrEP-eligible cis Black women through convenience sampling using electronic flyers (e-flyers) that were disseminated through social media channels of academic and community partners as well as in-person recruitment through social networks of community-based agencies and clinics. Eligible participants were assigned female sex at birth, were 18 years or older, had been sexually active with an opposite sex partner within the past 6 months, were fluent in English, and had access to a phone or internet. Cis Black women who were not able to physically provide informed consent due to a cognitive impairment or psychological distress, were unable or unwilling to meet study requirements, were ineligible for PrEP (i.e. known HIV positive), or were already on PrEP were not eligible to participate. E-flyers contained…’

Comment: In the third paragraph in the Materials and Methods section, the FGD and baseline survey are mentioned – are these the same thing?

Response: The baseline survey and the focus group discussion are not the same thing. In an effort to offer more clarity, further details are now provided to distinguish the purpose of the baseline survey from the discussion of the focus group discussion (FGD) guide. The description of the guide is now at the onset of a new paragraph, paragraph 4, as well. This section on page 3, paragraph 3, lines 133-135, now reads as,

‘Once the session was scheduled, participants completed a baseline survey in Qualtrics to provide their contact information (i.e. email, phone number) for use by the research coordinator to follow-up with them on the next business day to explain the details of study participation.

The FGD guide…’

Comment: The third paragraph in the materials and methods section also mentions Theory of Gender and Power and The Sexual Script – There should be a few cited sentences describing what these are.

Response: In response to this comment, a few sentences were added to the new fourth paragraph on page 3, lines 137-145, to describe the two theories. This section now reads as, ‘The FGD guide included open-ended questions about the influences that culture, race, and gender have on PrEP use with cis Black women. The theoretical framework to justify these influences on PrEP use are rooted in the Theory of Gender and Power and The Sexual Script theory. The Theory of Gender and Power describes how societal norms influence gender-based inequities in a culture where males are socialized and supported in controlling decisions, which inform relationship dynamics and the way partners communicate and make sexual decisions.11 The Sexual Script Theory explores the ways culture shapes perception and expressions of sexual behavior in ways that individuals and partners deem appropriate and socially acceptable. The script becomes the roadmap of how sexual cues are interpreted and the behavior that follows.11 The FGD guide explored elements…’

Comment: It seems like the focus groups were online as it was noted videoconferencing software was used – the manuscript should explicitly detail how the groups were conducted, in person or virtual.

Response: The focus groups were conducted using video conferencing. Paragraph 1 in the Materials and methods section, page 3, lines 111-112, has been revised to explicitly state that the focus groups were conducted in a virtual environment. Sentence 1 now states, ‘This is a qualitative observational study using focus group discussions in a virtual environment as the communication medium to explore factors that may influence PrEP initiation among a cohort of 20 cis Black women.’

Comment: The limitations section also noted a variance between sites – more description of the sites (all different facilitators? Research experience? Number of participants in each) would be helpful. The variance between sites should either be mentioned in the results and/or clarified/defined in the limitations – as it is now, it is not clear to what the variance is referring.

Response: This is an error and oversight on the part of the authorship team. This is a single site study. There was a single facilitator for all focus groups. There was a single focus group guide used across all focus groups. There variance in research experience is amongst coders, not sites. The limitations section on page 27, paragraph1, lines 929-931 has been revised and now accurately reads as, ‘The research experience level of focus group facilitators varied between coders.

Comment: Line 119 also mentions the next phase of the study for the first time – It would be helpful if the next phase was introduced at the end of the introduction, where it describes the current study as a necessary first step. Reference at line 119 then would have appropriate context.

Response: As the authorship team wrote a separate paper on PrEP readiness, we presume that this is an error. This study did not focus on PrEP readiness at all. This study focused on facilitators and barriers to the decision to initiate PrEP use among cisgender Black women. The sentence on page 3, the final paragraph of the introduction section, lines 103-104 has been revised to read, ‘As such, we led a qualitative study to explore facilitators and barriers that influence the decision for PrEP initiation with cis Black women.’

As suggested, this sentence was added at to the last paragraph of the introduction section on page 3, lines 104-109, Qualitative findings here informed the next phase of the research, which was the development of health communication material from the perspective of cis Black women to inform the content of a video-log based intervention to enhance access to PrEP, culminating in the final phase to pilot test the vlogs and assess whether they facilitated behavior change, specifically an increase in efforts to access PrEP over a brief follow-up period.’

Comment: Lines 121 and 122 mention the theory of planned behavior – a citation is needed here.

Response: A citation relative to the theory of planned behavior in the sexual decision-making process for Black women, identifying urban Black adolescents' beliefs about male-to-female verbal sexual coercion, was added to line 160 in the final paragraph of the materials and methods section before the data analysis subsection. The authorship team perceives that this ties in nicely to the presentation of material regarding the influence of one’s locus of control on sexual health decision-making.

Reference: Eaton AA, Stephens DP. Using the Theory of Planned Behavior to Examine Beliefs About Verbal Sexual Coercion Among Urban Black Adolescents. J Interpers Violence. 2016.

Comment: Line 128 looks to have a type – live instead of life

Response: Thank you for pointing this out. We have corrected this typo on line 167 in the final paragraph of the materials and methods section before the data analysis subsection.

Comment: Line 134 and 135 note specific counties – consider describing location rather than giving specific county names (rural, urban, southern, etc.)

Response: This is actually an error. Participants were from a single county. However, as recommended, the county names were removed and replaced with descriptors, specifically ‘the largest, most diverse, and populous county in the South’ in paragraph 1 of the section 2.1 Data Analysis, paragraph 1, lines 173-174.

Comment: Table 1 describes the participants who filled out the screening survey, yet only 20 participants were included in the focus groups. How did the team get from the 241 responses to only 20 participants? Consider providing demographics of the participants, possibly in comparison to the 241 responses to consider whether the 20 participants are representative of the original 241 responses.

Response: Demographic data from participants are now provided in Table 1 in comparison to the 241 screened responses.

Comment: Table 1 has only two choices for race – African American or Black – were any other choices provided? Also, there are different N for each variable – why? Did some participants only fill out some variables?

Response: As the study aimed to enroll cisgender Black women only, the screening aimed to include women who identified as African American or Black. All other racial identifiers were excluded from study participation.

Yes, the different Ns for each question represent the number of responses for those questions. There were no forced responses for this survey.

Comment: The sentence spanning lines 149 – 151 is unclear.

Response: The sentence reads, ‘A Kappa statistic was calculated based on all codes using IBM SPSS v26.0, a quantitative analytic software package, was used to assess inter-rater reliability among coding team members to assess and measure agreement.’ This sentence was revised and now states on page 5, paragraph 2 of the 2.1 Data Analysis subsection, lines 190-194, ‘Using the statistical software package IBM SPSS v 26.0, the team calculated a Kappa statistic to assess inter-rater agreement between coders.’ 

Comment: In the Results, it is noted that there are 1,217 codes; how many are among the highest frequency/listed in the table/included in descriptions vs. how many are left out?

Response: The team appreciates the inquiry. In an effort to depict a narrative that was most representative of the voices of the study population, the team decided to discuss more robustly the codes that were used most frequently in text while still mentioning all of the codes used in the tables. In several cases, a code was used only one time for a total of 16 times. Under each overarching theme, external and internal locus of control, each subtheme is discussed relative to the most frequently used codes. The top two most frequently used codes, and in rare cases the top three most frequently used codes, per theme, are described under each subtheme.

Comment: Also, how was it determined which codes would be described – there are multiple instances where codes with only 1 observation are described in the text and listed in the table.

Response: In an effort to depict a narrative that was most representative of the voices of the study population, the team decided to discuss more robustly the codes that were used most frequently in text while still mentioning all of the codes used in the tables. In several cases, a code was used only one time for a total of 16 times. Under each overarching theme, external and internal locus of control, each subtheme is discussed relative to the most frequently used codes. The top two most frequently used codes, and in rare cases the top three most frequently used codes, per theme, are described under each subtheme. In instances where codes with only one observation is described in the text, this is where a single code under a theme was ranked as one of the most frequently used code, which was the case for multiple internal codes (i.e. acceptance of PrEP as approved, awareness of HIV risk with drug use, and perceived HIV knowledge).

Comment: Consider revising formatting to remove bullets for the themes – continuation of numbering system would be easier to follow. Line 180, 3.11 External Locus of control, should be 3.1.1. Theme 1 in that section could then follow 3.1.1.1.

Response: The section numbers were changed throughout the results section for themes and codes per the recommendation of the reviewer.

RESULTS

Comment: The overall presentation of the Results would benefit from more summary – Themes are provided, but with no description. A few sentences describing what responses had in common in each theme, why reviewers grouped codes/responses in the given theme, etc., would be helpful.

Response: A brief description for each theme has now been provided.

Comment: Likewise, a summary of each code would be helpful rather than only the single highlighted quote, especially in cases where 20+ responses are included under 1 code. For example, lines 207 – 215 describe 1 example of media influence, but there are 60 uses of this code – a summary of the different points covered in those 60 uses, and possibly another direct quote or two, would be helpful. The quotes selected and analyses thereof were well done and easy to follow. Noting how many responses are covered in each code, and a summary of the different points covered, would provide more context for the results.

Response: A summary of each code and a summary of each theme is now provided throughout the results section. An additional quote, with a description, has been added to the code TV Media messaging about PrEP

In regards to noting how many responses are covered in each code, we have Tables to reflect this information and it would be redundant to share this information for each code. In cases where the codes were used at a high frequency, that information is now noted when deemed necessary by the authorship team.

Comment: Throughout description of the results, it is also noted frequently that the facilitator of a specific group asked a specific question – this gives the impression that not all groups received the same questions. Clarification is needed, and if all groups did not receive the same questions, that should be addressed.

Response: There was a single facilitator for all focus groups. There was a single focus group guide used across all focus groups.

In order to provide context for each quote, the writer of the results section went back to each transcript and provided context around each quote to frame the quote in the context of the discussion. This framing is not meant to imply that something was done differently in one group than in any other group.

Comment: Lines 306 – 311 describe cost related to PrEP – the quote reads like it is referring to personal cost, not financial, though is coded as financial cost – is there more context to the quote that led it to being classified as financial cost rather than a personal cost?

Response: The participant describes the cost, the risk, and the benefit. The team perceives that the ‘risk’ refers to the personal cost and the ‘benefit’ refers to the personal benefit. The cost refers to the financial cost.

Comment: Consistency should be used when referencing groups and participants – manuscript changes from numerals (ex. 3) to words (ex. Three).

Response: The manuscript has been reviewed and numerals are now used throughout, replacing the words for the numbers.

Comment: Line 393 is Reasons women might be hesitant towards taking PrEP – this seems like a different format than other codes, and the example quote provided looks like it could fit with other previously provided codes. Also, the title of the code implies that multiple disconnected responses could be grouped under this code. In the analysis of the quote, it notes there is a hesitancy in taking medication that is not needed, and 57% of the responses referred to vaccine hesitancy – it seems like these statements would better inform the name of the code. Additionally, noting that 57% of the responses referred to vaccine hesitancy is very helpful – statements like this would help the reader better understand the constellation of responses – this sort of summary is what is meant by summaries of themes/codes.

Response: This code, ‘reasons why women might be hesitant towards taking PrEP’ was used in eight instances. Vaccine hesitancy was a primary reason for hesitancy when considering PrEP use with the injection modality. As all reasons for hesitancy were not vaccine use, vaccine use did not inform the naming of the code. We agree that the statement regarding the percentage of individuals referring to vaccine hesitancy helped to inform the reader; however, this type of uniformity is not present across all codes and this type of grouping in such a succinct manner is not always readily available in qualitative research. As recommended by the reviewer, more summary level data is provided for themes.

Comment: Line 426 notes how quotes from two different participants coded in two different codes are similar – again, summaries of the codes would help the reader understand how they differ rather than leaving them wondering why they aren’t grouped together under the same code.

Response: In an effort to offer further clarity with how participant 2 (Code: Perception that PrEP offers safety) and 4 (Code: PrEP empowers ability to control personal sexual health) have similarities and substantial differences in perspectives, more context is provided relative to the description of the presentation of the perception of participant 2 relative to participant 3 regarding the code ‘Perception that PrEP offers safety’. This addition shows that although the initial code for participant 2 is accurate, the second code applies and is additive of the initial code. It now reads on page 17, under Code 3.1.2.2.3.2. lines 601-603, ‘        Similar to how participant 2 describes the feelings of safety that PrEP offers her, she also describes it as a source of empowerment to control her ability to prevent herself from contracting HIV, participant four perceives PrEP as a pathway to control what happens to one’s body.’

Comment: Line 551 has a possible type – addition rather than addiction.

Response: Thank you for pointing this out. We have corrected this typo on page 21, line 788.

Comment: Line 561 notes 2 code as the first code under theme 10

Response: The code numbers have been changed. Theme 10 is now coded as 3.1.2.2.10 and the code under theme 10 is numbered as 3.1.2.2.10.1.

Comment: Table 3 – perception that PrEP is for people who inject drugs is in twice

Response: One of the lines were deleted.

DISCUSSION

Comment: In the discussion, internal vs. external loci of control is the focus – did participants exhibit responses that indicated both internal vs. external loci of control, or did individuals skew either more internal vs. external? Assuming they most often displayed a mix of both, highlighting the need to address the internal locus of control via intervention makes sense; however, if a significant number of participants expressed only external locus of control in their response, such an intervention runs the risk of missing the needs of those who only exhibit an external locus of control.

Response: The unit of analysis for this study was the focus group instead of each individual participant. Thus, comparison of locus of control per participant is not possible in this paper; however, this, exploration of internal versus external locus of control at the individual-level for each participant, is an approach that we can consider for future papers.

Comment:         The sentence in lines 656 through 659 needs citation.

Response: Citation added and statement revised a bit and now reads on page 26, paragraph 5, lines 908-912, ‘Due to the inherent limitations of an external locus of control on the preservation of an HIV negative status amidst myriad determinants of health, HIV primary prevention strategies are more poised for short-term success if the outcome is focused on individual-level behaviors instead of expectations to overcome the external environment.49

Comment: The last paragraph in the limitations is only one sentence – it should either be combined to the previous paragraph or have additional statements added.

Response: We have combined the last sentence of the second paragraph in the previous paragraph on lines 947-948.

Reviewer 2 Report

Comments and Suggestions for Authors

This is a robust study on an important topic, improving PrEP uptake and awareness among black ciswomen. There are several areas in which this paper could be strengthened. The authors state in the introduction that they are assessing PrEP readiness - but not question on PrEP readiness appears and this concept is not in the results. There are several frameworks mentioned in the methods - it is unclear how they are selected or used; the results are presented as internal vs external loci but there is no information about how this relates to the earlier frameworks. A strong theoretical framework would greatly help with the results, which require a great degree of synthesis and stronger conclusions in general.

Introduction and abstract state different percentages (67% vs nearly 60^%) for proportion of new female cases among CBW.

No semicolon needed in line 35 - its one complete sentence.

Appreciate the authors focus on systems and structures

line 44 - I would pushback gently here - there is quite bit of literature on barriers to PrEP use and low PrEP awareness among CBW (1.

Hill SV, Pratt MC, Elopre L, Simpson T, Gaines Lanzi R, Matthews LT. “Nobody wants to have conversation about HIV.” A thematic analysis of in-depth interviews with Black adolescent women and providers about strategies for discussing sexual health and HIV prevention. Sexually Transmitted Diseases. 2024 Apr 11;10.1097/OLQ.0000000000001972.;  1. Hirschhorn LR, Brown RN, Friedman EE, Greene GJ, Bender A, Christeller C, Bouris A, Johnson AK, Pickett J, Modali L, Ridgway JP. Black Cisgender Women’s PrEP Knowledge, Attitudes, Preferences, and Experience in Chicago. J Acquir Immune Defic Syndr. 2020;84(5):11.; 1. Johnson AK, Fletcher FE, Ott E, Wishart M, Friedman EE, Terlikowski J, Haider S. Awareness and Intent to Use Pre-exposure Prophylaxis (PrEP) Among African American Women in a Family Planning Clinic. J Racial and Ethnic Health Disparities. 2020 Jun;7(3):550–554.; 1. Willie TC, Knight D, Baral SD, Chan PA, Kershaw T, Mayer KH, Stockman JK, Adimora AA, Monger M, Mena LA, Philllips KA, Nunn A. Where’s the “Everyday Black Woman”? An intersectional qualitative analysis of Black Women’s decision-making regarding HIV pre-exposure prophylaxis (PrEP) in Mississippi. BMC Public Health. 2022 Aug 23;22(1):1604.; 1. Irie WC, Calabrese SK, Mayer KH, Geng EH, Blackstock O, Marcus JL. Social and structural factors associated with interest in HIV preexposure prophylaxis among Black women in the United States. AIDS Care. Taylor & Francis; 2024;0(0):1–10. PMID: 38176016; 1. Irie WC, Calabrese SK, Mayer KH, Geng EH, Blackstock O, Marcus JL. Social and structural factors associated with interest in HIV preexposure prophylaxis among Black women in the United States. AIDS Care. Taylor & Francis; 2024;0(0):1–10. PMID: 38176016;  1. Flash CA, Stone VE, Mitty JA, Mimiaga MJ, Hall KT, Krakower D, Mayer KH. Perspectives on HIV Prevention Among Urban Black Women: A Potential Role for HIV Pre-Exposure Prophylaxis. AIDS Patient Care STDS. 2014 Dec 1;28(12):635–642. PMCID: PMC4250961; 1. Haider S, Friedman EE, Ott E, Moore A, Pandiani A, Desmarais C, Johnson AK. Knowledgeable, aware / interested: Young black women’s perceptions of pre-exposure prophylaxis. Front Reprod Health. 2022 Sep 30;4:671009. PMCID: PMC9580699; 1. Hill LM, Lightfoot AF, Riggins L, Golin CE. Awareness of and attitudes toward pre-exposure prophylaxis among African American women living in low-income neighborhoods in a Southeastern city. AIDS Care. 2020 May 25;1–5.; 1. Knight D, Saleem HT, Stockman JK, Willie TC. Experiences of Black Women in the United States Along the PrEP Care Continuum: A Scoping Review. AIDS Behav. 2023 Jul 1;27(7):2298–2316.) to mentiona few line 46 - again no semicolon required here. line 70 - I agree these needs are diverse and I'm not sure how the authors conclude that communicating information is the best approach or even an appropriate approach to deal with low income or substance use, for example. Again referring to structural barrieres, women may need other services in conjunction with PrEP Line 87 - is there a framework the authors are using to define PrEP readiness and PrEP initiation   Methods: I see the authors do use Theory of Gender and Power and Sexual Script Theory. Can the authors describe these theories and why they selected them? Also the Theory of Planned Behavior Table 1 - should this be of all screened participants or just those who participated? Can the authors include the interview guide as an appendix? What was the final sample size (abstract suggests 20)? How were they selected from the screened individuals? How many focus groups were there and how many participants each? Where participants HIV negative? If not, what was the reasoning?   Results: Don't need to include Codes. Overall, more synthesis across codes would be helpful. Currently, most of the text just summarizes the quotes. I'm unclear how preference for certain provider types is self-advocacy - nor how self-advocacy (by definition one's own ability to argue for oneself ) would not be an internal locus line 249 - it is unclear from the quote if the participant actually had more than one partner? In theme 4 it appears that participants can have very different reactions around providers asking to test for STIs. How do you account for this and what does it imply for messaging?    Why are personal concerns about injections an external locus? The division between external and internal locus is very confusing. There are examples of learning about PrEP from TV shows or commercials - and one is considered external and the other internal.             Comments on the Quality of English Language

Sufficient, minor grammatical errors. 

Author Response

Comment: The authors state in the introduction that they are assessing PrEP readiness - but not question on PrEP readiness appears and this concept is not in the results.

Response: As the authorship team wrote a separate paper on PrEP readiness, we presume that this is an error. This study did not focus on PrEP readiness at all. This study focused on facilitators and barriers to the decision to initiate PrEP use among cisgender Black women. The sentence on page 3, the final paragraph of the introduction section, lines 103-104 has been revised to read, ‘As such, we led a qualitative study to explore facilitators and barriers that influence the decision for PrEP initiation with cis Black women.’

Comment: There are several frameworks mentioned in the methods - it is unclear how they are selected or used;

Response: In response to this comment, more clarity is provided regarding the three frameworks discussed in this manuscript. In response to this comment, a few sentences were added to the new fourth paragraph on page 3, lines 137-145, to describe the two theories. This section now reads as, ‘The FGD guide included open-ended questions about the influences that culture, race, and gender have on PrEP use with cis Black women. The theoretical framework to justify these influences on PrEP use are rooted in the Theory of Gender and Power and The Sexual Script theory. The Theory of Gender and Power describes how societal norms influence gender-based inequities in a culture where males are socialized and supported in controlling decisions, which inform relationship dynamics and the way partners communicate and make sexual decisions.11 The Sexual Script Theory explores the ways culture shapes perception and expressions of sexual behavior in ways that individuals and partners deem appropriate and socially acceptable. The script becomes the roadmap of how sexual cues are interpreted and the behavior that follows.11 The FGD guide explored elements…’

A citation relative to the theory of planned behavior in the sexual decision-making process for Black women, identifying urban Black adolescents' beliefs about male-to-female verbal sexual coercion, was added to line 160. The authorship team perceives that this ties in nicely to the presentation of material regarding the influence of one’s locus of control on sexual health decision-making.

Reference: Eaton AA, Stephens DP. Using the Theory of Planned Behavior to Examine Beliefs About Verbal Sexual Coercion Among Urban Black Adolescents. J Interpers Violence. 2016.

Comment: the results are presented as internal vs external loci but there is no information about how this relates to the earlier frameworks.

Response: The frameworks describe how the focus group guide’s content was crafted in an effort to elicit data to inform facilitators and barriers to PrEP initiation among cisgender Black women. The internal and locus of control frameworks are used to stratify the perspectives of how cis Black women, at the focus group unit of analysis, described facilitators and barriers to PrEP initiation. The frameworks apply to separate aspects of the research:

Prior to data collection: Theory of Gender and Power, Sexual Script Theory, Theory of Planned Behavior

Post data collection (data synthesis, data analysis): Internal and External Locus of Control

Comment: A strong theoretical framework would greatly help with the results, which require a great degree of synthesis and stronger conclusions in general.

Response: The team perceives that the theoretical framework used, the psychological framework of a locus of control, is a strong theoretical framework to describe how and why the facilitators and barriers influence the decision for PrEP initiation among cisgender Black women. The team has made a significant commitment to address the comments of the reviewers to provider stronger presentation and conclusions in general.

Comment: Introduction and abstract state different percentages (67% vs nearly 60^%) for proportion of new female cases among CBW.

Response: Thank you for this comment. The data points in the abstract and in the introduction section are actually presenting two separate data points.

Cisgender Black women (CBW) experience 67% of new HIV diagnoses among women in the South. The last 3 words were added to this sentence in the abstract on lines 10-11.

Reference: Linley L, Johnson AS, Song R, et al. Estimated HIV incidence and prevalence in the United States, 2014–2018. 2020. [Google Scholar] [Ref list]

It is also factual that ‘Black women account for more than half (58%) of those infections (Gantz et al., 2018), despite representing only 12.9% of reproductive women in the US (US Census Bureau, 2020).’

References:

Gant Z, Johnson SD, Li J, et al. Diagnoses of HIV infection in the United States and dependent areas (updated), 2018. 2020.

United States Census Bureau .Annual estimates of the resident population by sex, race, and Hispanic origin. 2020.

This information was also referenced in:

Willie TC, Knight D, Baral SD, Chan PA, Kershaw T, Mayer KH, Stockman JK, Adimora AA, Monger M, Mena LA, Philllips KA, Nunn A. Where's the "Everyday Black Woman"? An intersectional qualitative analysis of Black Women's decision-making regarding HIV pre-exposure prophylaxis (PrEP) in Mississippi. BMC Public Health. 2022 Aug 23;22(1):1604. doi: 10.1186/s12889-022-13999-9. PMID: 35999528; PMCID: PMC9396836.

In response to this comment, the first sentence of the abstract has been revised to read, ‘Cisgender Black women (CBW) experience 67% of new HIV diagnoses among women in the South.’

Comment: No semicolon needed in line 35 - its one complete sentence.

Response: Semicolon deleted.

Comment: Appreciate the authors focus on systems and structures

Response: Thank you.

Comment: line 44 - I would pushback gently here - there is quite bit of literature on barriers to PrEP use and low PrEP awareness among CBW.

Nine references over the last ten years (2014-2024) were shared by the reviewer to mention a few.

Hill SV, Pratt MC, Elopre L, Simpson T, Gaines Lanzi R, Matthews LT. “Nobody wants to have conversation about HIV.” A thematic analysis of in-depth interviews with Black adolescent women and providers about strategies for discussing sexual health and HIV prevention. Sexually Transmitted Diseases. 2024 Apr 11;10.1097/OLQ.0000000000001972.;1.

Hirschhorn LR, Brown RN, Friedman EE, Greene GJ, Bender A, Christeller C, Bouris A, Johnson AK, Pickett J, Modali L, Ridgway JP. Black Cisgender Women’s PrEP Knowledge, Attitudes, Preferences, and Experience in Chicago. J Acquir Immune Defic Syndr. 2020;84(5):11.

Johnson AK, Fletcher FE, Ott E, Wishart M, Friedman EE, Terlikowski J, Haider S. Awareness and Intent to Use Pre-exposure Prophylaxis (PrEP) Among African American Women in a Family Planning Clinic. J Racial and Ethnic Health Disparities. 2020 Jun;7(3):550–554.

Willie TC, Knight D, Baral SD, Chan PA, Kershaw T, Mayer KH, Stockman JK, Adimora AA, Monger M, Mena LA, Philllips KA, Nunn A. Where’s the “Everyday Black Woman”? An intersectional qualitative analysis of Black Women’s decision-making regarding HIV pre-exposure prophylaxis (PrEP) in Mississippi. BMC Public Health. 2022 Aug 23;22(1):1604.

Irie WC, Calabrese SK, Mayer KH, Geng EH, Blackstock O, Marcus JL. Social and structural factors associated with interest in HIV preexposure prophylaxis among Black women in the United States. AIDS Care. Taylor & Francis; 2024;0(0):1–10. PMID: 38176016; 

Irie WC, Calabrese SK, Mayer KH, Geng EH, Blackstock O, Marcus JL. Social and structural factors associated with interest in HIV preexposure prophylaxis among Black women in the United States. AIDS Care. Taylor & Francis; 2024;0(0):1–10. PMID: 38176016 (DUPLICATE)

Flash CA, Stone VE, Mitty JA, Mimiaga MJ, Hall KT, Krakower D, Mayer KH. Perspectives on HIV Prevention Among Urban Black Women: A Potential Role for HIV Pre-Exposure Prophylaxis. AIDS Patient Care STDS. 2014 Dec 1;28(12):635–642. PMCID: PMC4250961; 

Haider S, Friedman EE, Ott E, Moore A, Pandiani A, Desmarais C, Johnson AK. Knowledgeable, aware / interested: Young black women’s perceptions of pre-exposure prophylaxis. Front Reprod Health. 2022 Sep 30;4:671009. PMCID: PMC9580699.

Hill LM, Lightfoot AF, Riggins L, Golin CE. Awareness of and attitudes toward pre-exposure prophylaxis among African American women living in low-income neighborhoods in a Southeastern city. AIDS Care. 2020 May 25;1–5.; 

Knight D, Saleem HT, Stockman JK, Willie TC. Experiences of Black Women in the United States Along the PrEP Care Continuum: A Scoping Review. AIDS Behav. 2023 Jul 1;27(7):2298–2316.)

Response: The team welcomes the push back and as such appropriately reframed the statement to read more accurately. It now reads as, ‘However, less than 2% of eligible cis Black women take PrEP.6 Healthcare providers, public health practitioners, and researchers are aware of the dire need to increase PrEP initiation with cis Black women; yet, the paucity of funded interventional research focused on effectively linking cis Black women to PrEP initiation persists.’

Comment: Quality of English language - Sufficient, minor grammatical errors. 

Response: Grammatical errors are not identified; however, after addressing all 3 reviewer comments, authors will do a final review for grammar prior to submission of revisions.

Comment: line 46 - again no semicolon required here.

Response: Semicolon deleted.

Comment: line 70 - I agree these needs are diverse and I'm not sure how the authors conclude that communicating information is the best approach or even an appropriate approach to deal with low income or substance use, for example. Again referring to structural barriers, women may need other services in conjunction with PrEP

Response: Further evidence has been added to better link the connection of addressing the diversity of needs through health communication on behalf of Black women as an appropriate approach, based on historical evidence. The updated sentence in paragraph 4 of the introduction section, page 2, lines 83-87, reads as, ‘The facilitators and barriers to PrEP noted here are diverse. It has been posited that the use of vlogs for sexual health communication with cis Black women can be designed and implemented in innovative intervention strategies that connect with this audient in a way that aligns with the communication norms of society and communities.12  Consequently, approaches to increase PrEP initiation

Yes, the team agrees that it is plausible that further services in conjunction with PrEP may be needed for women; however, that premise is beyond the scope of the qualitative research presented here.

Reference: Hill MJ, Coker S. Novel Use of Video Logs to Deliver Educational Interventions to Black Women for Disease Prevention. West J Emerg Med. 2022 Feb 28;23(2):211-221. doi: 10.5811/westjem.2021.12.54012. PMID: 35302455; PMCID: PMC8967448.

Comment: Line 87 - is there a framework the authors are using to define PrEP readiness and PrEP initiation  

Response: As the authorship team wrote a separate paper on PrEP readiness, we presume that this is an error. This study did not focus on PrEP readiness at all. This study focused on facilitators and barriers to the decision to initiate PrEP use among cisgender Black women. The sentence on page 3, the final paragraph of the introduction section, lines 103-104 has been revised to read, ‘As such, we led a qualitative study to explore facilitators and barriers that influence the decision for PrEP initiation with cis Black women.’

PrEP initiation is now defined at first use in sentence one, defined as accepting an initial PrEP prescription in paragraph 1 of the introduction section, lines 28-29.

Reference: Joseph Davey DL, Mvududu R, Mashele N, Lesosky M, Khadka N, Bekker LG, Gorbach P, Coates TJ, Myer L. Early pre-exposure prophylaxis (PrEP) initiation and continuation among pregnant and postpartum women in antenatal care in Cape Town, South Africa. J Int AIDS Soc. 2022 Feb;25(2):e25866. doi: 10.1002/jia2.25866. PMID: 35138678; PMCID: PMC8826542.

 METHODS

Comment: I see the authors do use Theory of Gender and Power and Sexual Script Theory. Can the authors describe these theories and why they selected them? Also the Theory of Planned Behavior

Response: In response to this comment, a few sentences were added to the third paragraph of the Methods and Materials section, page 3, lines 137-145, to describe the two theories. This section now reads as, ‘The FGD guide included open-ended questions about the influences that culture, race, and gender have on PrEP use with cis Black women. The theoretical framework to justify these influences on PrEP use are rooted in the Theory of Gender and Power and The Sexual Script theory. The Theory of Gender and Power describes how societal norms influence gender-based inequities in a culture where males are socialized and supported in controlling decisions, which inform relationship dynamics and the way partners communicate and make sexual decisions.11 The Sexual Script Theory explores the ways culture shapes perception and expressions of sexual behavior in ways that individuals and partners deem appropriate and socially acceptable. The script becomes the roadmap of how sexual cues are interpreted and the behavior that follows.11 The FGD guide explored elements…’

A citation relative to the theory of planned behavior in the sexual decision-making process for Black women, identifying urban Black adolescents' beliefs about male-to-female verbal sexual coercion, was added to line 160. The authorship team perceives that this ties in nicely to the presentation of material regarding the influence of one’s locus of control on sexual health decision-making.

Reference: Eaton AA, Stephens DP. Using the Theory of Planned Behavior to Examine Beliefs About Verbal Sexual Coercion Among Urban Black Adolescents. J Interpers Violence. 2016.

Comment: Table 1 - should this be of all screened participants or just those who participated?

Response: Demographic data from participants are now provided in Table 1 in comparison to the 241 screened responses.

Comment: Can the authors include the interview guide as an appendix?

Response: Yes, the focus group guide is now included as an Appendix.

Comment: What was the final sample size (abstract suggests 20)?

Response: The final sample size was 20. Table 1 has been updated to reflect this update.

Comment: How were they selected from the screened individuals?

Response: Screened individuals were selected for study participation based on how their responses aligned with the study eligibility criteria. Screening individuals who met the eligibility criteria were called by the research coordinator on the next business day, debriefed about the study, and given the consent form via DocuSign. Upon receipt of the signed consent form, they were scheduled for a focus group discussion. Individuals who showed up on the day of the focus group discussion and participated were enrolled as participants. This process is described in paragraphs 1-3 of the Materials and Methods section of the paper.

Comment: How many focus groups were there and how many participants each?

Response: A total of 7 focus groups. Page 4, paragraph 4 of the materials and methods section, line 154 has been updated to read, ‘The study team conducted a total of 7 focus groups, coded 7 FGD transcripts, compared codes,…’

Comment: Were participants HIV negative? If not, what was the reasoning?  

Response: PrEP-eligibility was a part of the inclusion criteria; and all participants were asked about their HIV status, via self-report, during the screening process. Based on self-report, all participants were HIV negative.

RESULTS

Comment: Don't need to include Codes. Overall, more synthesis across codes would be helpful. Currently, most of the text just summarizes the quotes.

Response: More synthesis across codes is provided throughout the manuscript.

Comment: I'm unclear how preference for certain provider types is self-advocacy - nor how self-advocacy (by definition one's own ability to argue for oneself) would not be an internal locus line 249 - it is unclear from the quote if the participant actually had more than one partner?

Response: In the quote from participant 2 from focus group 3, she shared with her provider that she had one partner. This was not a truth, as she describes ‘if I would have been honest’. If she had been honest, she would have shared with her provider that she was having sex with several individuals and that she was having condomless sex. She describes that the conversation would have been a greater discussion, as she perceived the provider as being ‘open. The code for this quote is ‘Communicating openly with your provider is part of your health’ under the theme of ‘Comfort with provider-patient interactions’.

Based on my summation with the qualitative data, preference for certain provider types is connected to self-advocacy because the comfort with that provider creates the space for self-advocacy. It seems as though when cis Black female participants felt comfortable communicating, they felt comfortable being honest, speaking their mind, and engaging in a conversation as their full selves, which seemed to empower self-advocacy. The external locus of control here is the healthcare provider who either possess those qualities or creates that environment or not. When those elements are absent in the external environment, it appeared as though the external environment failed to catalyzed the internal locus of control to spark self-advocacy, a quality that is harnessed and controlled by the internal locus of control. (this paragraph was added in between the external and internal locus of control paragraph of the paper)

It is important to refrain from engaging in interpretation of the results during the results sections. Thus, this type of description is inappropriate for discussion section. In an effort to add more context, without discussing, the text has been revised in this way under the code:

Page 9, line 339-341: ‘3.1.1.3.2. Code. Communicating openly with your provider is a part of your health. The ability to communicate openly with a healthcare provider was described was a response to the emotional safety that some cis Black women described when certain preferred provider characteristics were present’

During focus group 3, participant 2 shared a conversation that she had with a provider where she describes a link between honesty and perceiving the provider as open to communication.’

Comment: In theme 4 it appears that participants can have very different reactions around providers asking to test for STIs. How do you account for this and what does it imply for messaging?   

Response: As the lead author, I account for this by what I know to be true based on 22 years of working and living with and as part of this population of women. Cis Black women are not monolithic. There is a certain level of tailoring that will always be needed with behavioral interventions, even when homogeneity is present at the race, biological sex, gender, sexual orientation, etc. levels.

As this is a qualitative research study exploring the perceptions of cis Black women regarding the facilitators and barriers to PrEP initiation among them, accounting for diversity of responses around providers and the implications of those messages is beyond the scope of the research presented here.

Comment: Why are personal concerns about injections an external locus?

Response: The content of the quote for the code ‘concerns for PrEP injectable’ is what qualifies it for categorization under external locus of control. The participant links the personal concern of the PrEP injection to the history of medical malpractice to Black people being given injections. The concern isn’t the injection itself. It is the history of social injustice to people who share her race classification in this country in history that spurs the concern. This is an external factor and history that fuels her fear of recurring social injustice from external parties.

Comment: The division between external and internal locus is very confusing.

Response: The addition of context around each theme hopefully at further clarity around the division between external and internal locus of control. In addition, the sentence between the paragraphs in the discussion section between external and internal locus of control hopefully adds further clarity.

Comment: There are examples of learning about PrEP from TV shows or commercials - and one is considered external and the other internal.            

Response: Table 7 describes all themes and codes for codes and themes associated with an internal locus of control. There are no references to media influences.

Reviewer 3 Report

Comments and Suggestions for Authors

I think that the results can be reproduced with other communities

1.- WHAT ARE THE SOCIAL AND PERSONAL FACTORS THAT INFLUENCE IN THE DECISION MAKING TO INITIATE HIV PRE-EXPOSURE PROPHYLAXIS IN BLACK WOMEN?

2.- IT IS INTERESTING HOW HUMAN ACTION IS AN UNKNOWN SINCE THE CONSTANT CAMPAIGNS ON AWARENESS AND VULNERABILITY CONTINUE WITHOUT HAVING A DEEP IMPACT ON SOME POPULATIONS THAT CONSCIOUSLY OR UNCONSCIOUSLY DO NOT ADMIT THE START OF GOOD PROPHYLAXIS

3.- IN MY OPINION IT IS AN HONEST WORK BECAUSE IT SHOWS IN TEXTUAL FORM THE RESPONSES OF THE INTERVIEWEES AND THESE ARE SO DISCONCERNING THAT THEY FORCE US TO THINK ABOUT NOT GIVING IN UNTIL WE ACHIEVE GLOBAL AWARENESS.

4.- I HAVE NO PROPOSAL REGARDING THE METHODOLOGY

5.- IT IS URGENT FOR THE AREA OF PUBLIC HEALTH TO TAKE INTO CONSIDERATION THAT GLOBAL CONSCIOUSNESS STILL DOES NOT EXIST IN MANY ETHNIC GROUPS AND THIS IS WHERE THEY SHOULD WORK HARDEST AND THIS IS DESCRIBED VERY WELL BY THE AUTHORS

6.- IT IS URGENT FOR THE AREA OF PUBLIC HEALTH TO TAKE INTO CONSIDERATION THAT GLOBAL CONSCIOUSNESS STILL DOES NOT EXIST IN MANY ETHNIC GROUPS AND THIS IS WHERE THEY SHOULD WORK HARDEST AND THIS IS DESCRIBED VERY WELL BY THE AUTHORS

7.- PARTICULARLY TABLES AND FIGURES SEEM CLEAR TO ME

Author Response

Comment: WHAT ARE THE SOCIAL AND PERSONAL FACTORS THAT INFLUENCE IN THE DECISION MAKING TO INITIATE HIV PRE-EXPOSURE PROPHYLAXIS IN BLACK WOMEN?

Response: We have added two sentences with a corresponding citation that contextualizes the social and structural discriminations that impede HIV prevention efforts for cis Black women on page 1, paragraph 1 of the introduction section, lines 39-42.

In addition, theoretical frameworks help to explain how social and personal factors influence the decision making to initiate HIV PrEP among cis Black women. A few sentences were added to the new fourth paragraph of the Materials and Methods section on page 3, lines 137-145, to describe two theories that are relevant to social and personal factors that influence sexual decision making among cis Black women. This section now reads as, ‘The FGD guide included open-ended questions about the influences that culture, race, and gender have on PrEP use with cis Black women. The theoretical framework to justify these influences on PrEP use are rooted in the Theory of Gender and Power and The Sexual Script theory. The Theory of Gender and Power describes how societal norms influence gender-based inequities in a culture where males are socialized and supported in controlling decisions, which inform relationship dynamics and the way partners communicate and make sexual decisions.11 The Sexual Script Theory explores the ways culture shapes perception and expressions of sexual behavior in ways that individuals and partners deem appropriate and socially acceptable. The script becomes the roadmap of how sexual cues are interpreted and the behavior that follows.11 The FGD guide explored elements…’

Comment: IT IS INTERESTING HOW HUMAN ACTION IS AN UNKNOWN SINCE THE CONSTANT CAMPAIGNS ON AWARENESS AND VULNERABILITY CONTINUE WITHOUT HAVING A DEEP IMPACT ON SOME POPULATIONS THAT CONSCIOUSLY OR UNCONSCIOUSLY DO NOT ADMIT THE START OF GOOD PROPHYLAXIS

Response: While we appreciate the comment of reviewer 3, this comment does not warrant a response or corrective action of the authorship team.

Comment: IN MY OPINION IT IS AN HONEST WORK BECAUSE IT SHOWS IN TEXTUAL FORM THE RESPONSES OF THE INTERVIEWEES AND THESE ARE SO DISCONCERNING THAT THEY FORCE US TO THINK ABOUT NOT GIVING IN UNTIL WE ACHIEVE GLOBAL AWARENESS.

Response: While we appreciate the comment of reviewer 3, this comment does not warrant a response or corrective action of the authorship team.

Comment: I HAVE NO PROPOSAL REGARDING THE METHODOLOGY

Response: While we appreciate the comment of reviewer 3, this comment does not warrant a response or corrective action of the authorship team.

Comment: IT IS URGENT FOR THE AREA OF PUBLIC HEALTH TO TAKE INTO CONSIDERATION THAT GLOBAL CONSCIOUSNESS STILL DOES NOT EXIST IN MANY ETHNIC GROUPS AND THIS IS WHERE THEY SHOULD WORK HARDEST AND THIS IS DESCRIBED VERY WELL BY THE AUTHORS

Response: While we appreciate the comment of reviewer 3, this comment does not warrant a response or corrective action of the authorship team.

Comment: PARTICULARLY TABLES AND FIGURES SEEM CLEAR TO ME

Response: While we appreciate the comment of reviewer 3, this comment does not warrant a response or corrective action of the authorship team.

Round 2

Reviewer 2 Report

Comments and Suggestions for Authors

Thank you to the authors for the thoughtful responses and effort that went into the revised draft. I have one remaining question as the external/internal loci is still confusing as an organizing concept. The authors describe loci of control as a personality trait. Were participants then screened or tested for thei loci of control and that is how the interviews were grouped and coded? Or could quotes from the same individual appear as both external and internal loci?

Author Response

Comment: Thank you to the authors for the thoughtful responses and effort that went into the revised draft.

Response: The authors appreciate the note of validation given the effort required to adequately respond to the comments.

Comment: I have one remaining question as the external/internal loci is still confusing as an organizing concept. The authors describe loci of control as a personality trait. Were participants then screened or tested for their loci of control and that is how the interviews were grouped and coded?

Response: Locus of control is a theoretical constructed derived from Julian B. Rotter (1954) in his social learning theory. The psychological definition of a locus of control refers to the degree in which an individual feels a sense of agency in regard to his or her life. The locus of control can be used to describe a person’s perception of what causes events in their life. Botha and Dahmann (2023) provide empirical evidence on a direct link between locus of control and self-control and how they interact to explain a range of health outcomes. Here, in this paper, we organize themes by locus of control and stratify them based on the authorship team’s purview of whether the factor influencing PrEP initiation is derived from an external or internal locus of control. The team then explains those associations with the decision to take PrEP as a preventive health outcome.

In regards to the use of the concept of the locus of control relative to participants, this was not done at the individual level. In the initial response to reviewers, for reviewer 1, the team made a concerted effort to articulate this approach to the analysis when stating, ‘The unit of analysis for this study was the focus group. The unit of analysis was not the individual participant. Thus, comparison of locus of control per participant is not possible in this paper; however, this, exploration of internal versus external locus of control at the individual-level for each participant, is an approach that we can consider for future papers.’

Participants were not screened or tested for their locus of control.

Prior to data collection, the theoretical frameworks were: Theory of Gender and Power, Sexual Script Theory, Theory of Planned Behavior

Post data collection (data synthesis, data analysis), the theoretical construct is: Internal and External Locus of Control

The concept of a locus of control is rooted in the field of psychology. The authorship team is adding a citation in hopes that it would bring clarity to the reviewer:

Reference: Botha F, Dahmann SC. Locus of control, self-control, and health outcomes. SSM Popul Health. 2023 Nov 24;25:101566. doi: 10.1016/j.ssmph.2023.101566. PMID: 38077246; PMCID: PMC10698268.

Comment: Or could quotes from the same individual appear as both external and internal loci?

Response: Based on the theme and framing, it is possible that quotes from the same individual could appear as both an external and internal locus of control. The added text under themes and codes is meant to offer clarity regarding distinguishing between both frames.